# Aerocyte specification and lung adaptation to breathing is dependent on alternative splicing changes

Marta F Fidalgo[1], Catarina G Fonseca[1], Paulo Caldas[2], Alexandre ASF Raposo[1], Tania Balboni[3], Lenka Henao-Mišíková[1], Ana R Grosso[2], Francisca F Vasconcelos[1,*], Cláudio A Franco[1,4,5,*]

Adaptation to breathing is a critical step in lung function and it is crucial for organismal survival. Alveoli are the lung gas exchange units and their development, from late embryonic to early postnatal stages, requires feedbacks between multiple cell types. However, how the crosstalk between the alveolar cell types is modulated to anticipate lung adaptation to breathing is still unclear. Here, we uncovered a synchronous alternative splicing switch in multiple genes in the developing mouse lungs at the transition to birth, and we identified hnRNP A1, Cpeb4, and Elavl2/ HuB as putative splicing regulators of this transition. Notably, we found that *Vegfa* switches from the *Vegfa* 164 isoform to the longer *Vegfa* 188 isoform exclusively in lung alveolar epithelial AT1 cells. Functional analysis revealed that VEGFA 188 (and not VEGFA 164) drives the specification of Car4-positive aerocytes, a subtype of alveolar endothelial cells specialized in gas exchanges. Our results reveal that the cell type–specific regulation of *Vegfa* alternative splicing just before birth modulates the epithelial-endothelial crosstalk in the developing alveoli to promote lung adaptation to breathing.

# Introduction

Respiration, the process of gas exchanges between the body and the environment, takes place at alveoli in the lung. The efficiency of gas exchanges is ensured by the functional specialization of the numerous cell types that compose the alveoli, including epithelial alveolar type 1 (AT1) cells, thin and elongated cells that line the surface of each alveolus; epithelial alveolar type 2 (AT2) cells, which are sparsely distributed at the alveolar surface and secrete surfactant essential for alveolar inflation and deflation; and endothelial cells (ECs), which compose the interior surface of capillaries that tightly enwrap the alveolar epithelial layer, and promote gas

exchanges with blood (Hogan et al, 2014; Mammoto & Mammoto, 2019).

Alveologenesis starts during late embryonic development at E16.5 in mouse and continues for 3–8 wk after birth. New alveoli form through branching morphogenesis and angiogenesis by co-ordinated growth and specialization of epithelial and ECs, respectively (Hogan et al, 2014; Vila Ellis & Chen, 2021). The crosstalk between these cell types is mediated by multiple signaling pathways, such as VEGFA, WNT, FGF, and HIPPO (Ellis et al, 2020; Vila Ellis & Chen, 2021; Kina et al, 2021; Zepp et al, 2021). For instance, secretion of VEGFA by epithelial cells during lung development regulates the expansion of the vascular network by binding to VEGF receptor 2 (VEGFR2) at the surface of ECs and triggering angiogenesis. Genetic deletion or pharmacological inhibition of VEGFA compromises lung alveolar epithelial development and capillary growth, leading to bronchopulmonary dysplasia, characterized by simplified alveoli and dysmorphic vasculature (Thébaud et al, 2005; Yamamoto et al, 2007; Ellis et al, 2020). Despite extensive research on lung development, how the communication between these cell types is modulated at the critical transition between embryonic to postnatal development is not fully understood.

Transcriptional and alternative splicing (AS) changes have previously been implicated in the regulation of multiple developmental processes (Baralle & Giudice, 2017; Brinegar et al, 2017; Weyn-Vanhentenryck et al, 2018; Farini et al, 2020). Previously published RNAseq and single-cell RNAseq studies from lungs at different developmental stages have enabled the comprehensive characterization of cell populations and the detailed study of gene expression changes occurring during lung development (Treutlein et al, 2014; Wang et al, 2018; Ellis et al, 2020). Yet, because of the lack of sequencing depth, none of these approaches has allowed the study of AS, and thus, the knowledge regarding AS during lung development is limited. One of the few genes shown to undergo AS changes during lung development was *Vegfa* (Healy et al, 2000; Ng et al, 2001; Greenberg et al, 2002). *Vegfa* is composed of eight exons and the most common *Vegfa* isoforms differ on the inclusion or

[1]Instituto de Medicina Molecular João Lobo Antunes, Faculdade de Medicina, Universidade de Lisboa, Lisboa, Portugal   [2]Department of Life Sciences, UCIBIO – Applied Molecular Biosciences Unit, NOVA School of Science and Technology, NOVA University Lisbon, Caparica, Portugal   [3]Department of Experimental, Diagnostic and Specialty Medicine, University of Bologna, Bologna, Italy   [4]Instituto de Histologia e Biologia do Desenvolvimento, Faculdade de Medicina, Universidade de Lisboa, Lisboa, Portugal   [5]Universidade Católica Portuguesa, Católica Medical School, Católica Biomedical Research Centre, Lisboa, Portugal

Correspondence: fvasconcelos@medicina.ulisboa.pt; cfranco@medicina.ulisboa.pt
*Francisca F Vasconcelos and Cláudio A Franco are co-last authors.

exclusion of exons 6 and 7: *Vegfa 188* contains all eight *Vegfa* exons, *Vegfa 164* does not contain exon 6, and *Vegfa 120* does not contain exons 6 and 7. *Vegfa* isoforms are functionally distinct in terms of binding to the extracellular matrix, and in their potential to induce angiogenesis, EC proliferation, survival and vascular permeability (Domigan et al, 2015; Yamamoto et al, 2016; Peach et al, 2018b; Bowler & Oltean, 2019). In addition to these classical isoforms, *Vegfaxxxb* isoforms have been identified and differ from the prior ones by a differential alternative splicing (DAS) of exon 8 (inclusion of exon 8b instead of 8a). *Vegfaxxxb* isoforms have been originally identified in human renal cell carcinoma and have been shown to have an anti-angiogenic potential (Bates et al, 2002). However, the existence of these isoforms in physiological contexts remains controversial (Harris et al, 2012; Bridgett et al, 2017; Dardente et al, 2020). During lung development, it has been shown that the relative proportion between *Vegfa 120*, *164*, and *188* isoforms changes (Healy et al, 2000; Ng et al, 2001; Greenberg et al, 2002) and that loss of specific isoforms has a functional impact. Mice expressing exclusively *Vegfa 120* showed impaired lung development, whereas mice expressing exclusively *Vegfa 164* or *Vegfa 188* isoforms have no gross morphological defects (Compernolle et al, 2002; Galambos et al, 2002). Despite the relevance of *Vegfa* AS during lung development, the temporal dynamics and cell type-specific expression of *Vegfa* isoforms along lung development remain poorly defined.

Here, we performed an unbiased genome-wide analysis of AS at late embryonic and early postnatal stages of lung development and we identified that most of the AS changes occur at the transition from the pre- to postnatal life, suggesting that AS regulation may be associated with lung adaptation to birth. We validated our genome-wide analysis by focusing on *Vegfa* isoforms and we identified a cell type–specific switch in AS of *Vegfa* during lung alveologenesis.

## Results

### Genome-wide analysis reveals that AS changes occur at the transition from the pre- to post-natal period

To analyze the genome-wide AS changes occurring during lung development, we extracted and sequenced mRNA of whole mouse lungs at two embryonic (E15.5 and E18.5) and two postnatal (P5 and P8) stages (Fig 1A). We performed 101 nt paired-end RNAseq with triplicate samples and obtained an average of 59.8 million reads per sample (Fig S1A).

To assess the quality of our datasets, we analysed the expression of a set of known lineage-specific markers (Fig S1B). In accordance with what has been shown before (Beauchemin et al, 2016; Ardini-Poleske et al, 2017), our data show that the epithelial progenitor marker *Sox9* progressively decreases, whereas markers for AT1 (*Aqp5* and *Pdpn*) and AT2 (*Sfptc* and *Sfptd*) epithelial cells increase over the same developmental period. Also, the levels of EC-specific markers—*Sox17*, *Pecam1*, and *Cdh5*—increase during development, as the lung progressively becomes more enriched in blood vessels. In addition, *Car4* expression increases from E18.5 to P5, concordant with previous findings showing that Car4-positive ECs, a specialized type of alveolar capillary ECs (aCap), is specified just before birth at E19.5

(Ellis et al, 2020). It has also been shown that, after birth, the immune cells repertoire that populates the lungs undergoes profound changes: embryonic macrophages (*Ncapd2*-positive) are eliminated and postnatal lungs become enriched in T lymphocytes (*Cd3e*-positive), neutrophils (Retnig-positive) and several subtypes of macrophages (*Itgax*-, *C1qa*-, and *Plac8*-positive) (Domingo-Gonzalez et al, 2020). Concordantly, in our datasets, the expression of *Ncapd2* decreases at birth, whereas the expression of all these other immune cell markers increases at birth. Of note, all these expression changes on lineage-specific markers are concordant with those identified in previously published datasets of gene expression during mouse lung development (LungMAP Consortium, www.lungmap.net and http://lungdevelopment.jax.org/) (Beauchemin et al, 2016; Guo et al, 2019). Thus, this analysis supports that the obtained RNAseq datasets are of good quality.

To identify AS events that change during lung development, we used the Vertebrate AS database (VastDB) and Vertebrate AS Transcription Tools (VAST-TOOLS) (Irimia et al, 2014; Tapial et al, 2017). We evaluated DAS between each pair of the above-mentioned perinatal time-points. We obtained 460 DAS events associated with 355 genes (Table S1) that showed a statistically significant variation in at least one of the pairwise comparisons (|ΔPSI| ≥ 10%, confidence interval of 95%). Because variations in gene expression levels can affect the accuracy with which AS can be detected and quantified (Gardina et al, 2006), we excluded from subsequent analysis all AS events associated with genes that undergo statistically significant changes in gene expression ($\log_2$FC > 1 and FDR < 0.05) in the same pairwise comparison (Table S2). From this filtering, we obtained 371 DAS events associated with 295 genes (Table S3). Remarkably, only a minority of genes associated with DAS events undergoes changes in gene expression in the same comparison (Fig 1B and Table S4). In addition, we found that most of the genes that undergo DAS are expressed at high levels (80%), as compared with the distribution of all expressed genes (~50%) (Fig 1C). Thus, these results suggest that most alterations in AS are independent of changes in gene expression levels of those same genes. To validate these AS changes identified in our RNAseq datasets, we performed RT-PCR on a subset of these events using independent RNA samples from mouse lungs at E18.5 and P5 (Fig S1C). We were able to validate 18/20 of the AS changes analysed (on two of them only one splicing variant could be detected) and found a strong correlation between the ΔPSI values obtained by the two methods (Fig S1D).

Next, we explored the dynamics of DAS during lung development. We performed K-means clustering of DAS events that occur in non-differentially expressed genes (Fig 1D), specifying the optimal number of clusters to 2, as determined by the average silhouette width method (Fig S1E). Remarkably, clustering of AS events segregated them between embryonic and postnatal stages. Cluster 1 contains AS events whose percent spliced in (PSI) values decrease postnatally, and cluster 2 contains AS events whose PSI values increase postnatally (Fig 1D and E and Table S5), with most of the AS changes detected occurring between E18.5 and P5. A complementary analysis using hierarchical clustering of these DAS events identifies this same trend (Fig S1F). This result suggests that adaptation to breathing (embryonic-to-postnatal transition) involves a comprehensive program of AS changes on a large number of

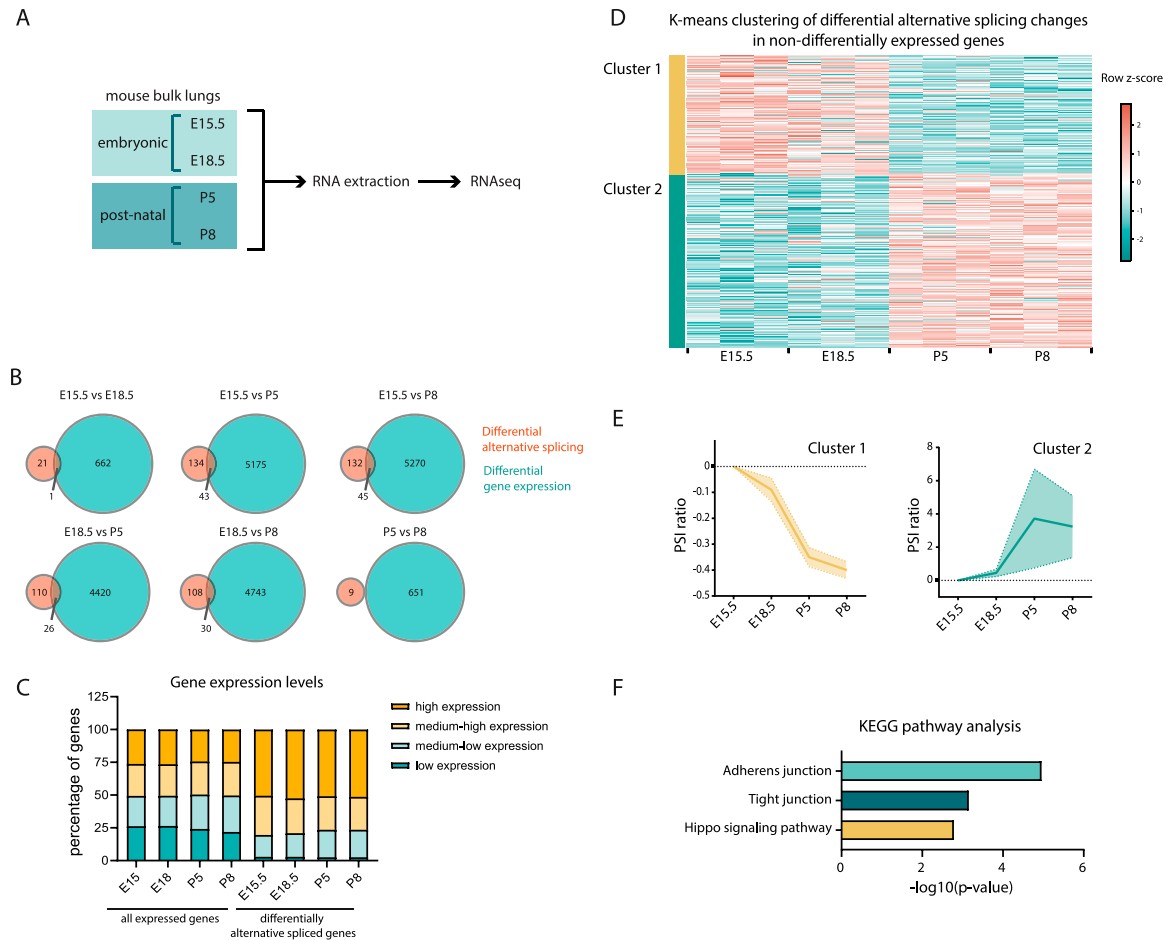

**Figure 1. Genome-wide analysis of alternative splicing (AS) during lung development reveals that most splicing changes occur at the perinatal period.**
**(A)** Schematic representation of workflow: Mouse lungs were collected at different developmental time-points. RNA was extracted and gene expression and splicing analyses were performed by RNAseq. **(B)** Venn diagrams representing the genes undergoing differential alternative splicing and/or differential gene expression in each pair-wise comparison between time-points. **(C)** Distribution of gene expression levels of genes undergoing differential AS in at least one pair-wise comparison between time-points. The 16,152 expressed genes expressed in the mouse developing lungs were grouped in four bins of equal size according to their absolute level of expression (CPMs) (low expression, medium-low expression, medium–high expression, and high expression). Then, the genes undergoing differential AS in at least one pairwise comparison between time-points were distributed within these bins. **(D)** Heat map representing the K-means clustering of differentially AS events in at least one pair-wise comparison between time-points associated with genes that do not undergo differential gene expression in the same comparison (K = 2). Cyan and coral represent decreased and increased percent spliced in (PSI), respectively, relative to the mean of each AS event across the time course (row z-score calculated from logit [PSI]). Table S5 contains all AS events associated with each cluster. **(E)** AS dynamics of the two kinetic clusters represented by mean ratio (PSI) (bold line) and 95% confidence interval (shaded area), data were centered in zero. **(F)** Enrichment of KEGG terms associated with differential alternative splicing events associated with non-differentially expressed genes. Table S6 contains the genes associated with each KEGG term and associated statistics.

genes, which might fine-tune their activity in a gene expression-independent manner.

To understand which pathways are associated with changes in AS during the embryonic-to-postnatal transition, we performed KEGG pathway analysis on the DAS events occurring in non-differentially expressed genes. Enriched terms reveal that AS occurs in genes that are associated with adherens junctions (such as *Afdn*, *Ctnnd1*, and *Baiap2*), tight junctions (such as *Magi1*, *Patj*, and *Amotl1*), and HIPPO signaling pathway (such as *Yap1*, *Tead1*, and *Llgl2*) (Fig 1F and Table S6). In sum, our results show that significant AS changes occur in the developing mouse lung during the embryonic-to-postnatal transition, between E18.5 and P5. These changes occur in genes involved in cell–cell adhesion complexes and a signaling pathway known to mediate intercellular communication during lung development.

## In silico analysis identifies RNA-binding protein (RBP) candidates for regulation of AS during lung development

The fact that most DAS occur between E18.5 and P5 suggests that these AS events may be regulated by a common regulatory mechanism. AS changes are often driven by changes in the levels of (RBPs) (Grosso et al, 2008; Baralle & Giudice, 2017). Thus, we sought to identify RBPs that could regulate the AS changes in lungs. We started by searching for motif enrichment/depletion on DAS events between E18.5 and P5 occurring in non-differentially expressed genes. We tested 250-nt sequences flanking the splice sites of all regulated splicing events (intronic and exonic) against all RBPs in the CISBP-RNA database (Ray et al, 2013) using Matt (Gohr & Irimia, 2019). For exon skipping events, we found a significant result (either

enrichment or depletion) for 49 motifs, corresponding to 39 RBPs, whereas for intron retention events, we found 64 motifs corresponding to 45 RBPs (Fig 2A and Table S7). We then focused on RBPs that change their expression during development. From the 363 mouse RBPs listed on the CISBP-RNA database, we filtered those that undergo differential gene expression between E18.5 and P5. We identified 47 RBPs fulfilling this condition, 11 of which increased in expression and 36 decreased in expression from E18.5 to P5 (Fig 2B and Table S8). Remarkably, only 3 of 47 RBPs exhibit both a significant enrichment of their motifs and differential gene expression between E18.5 and P5: *Hnrnpa1* (down-regulated), *Cpeb4* (up-regulated), and *Elavl2* (down-regulated) (Fig 2C and D and Table S9). Our results show that included exons have more motifs for the up-regulated *Cpeb4* and for the down-regulated *Elavl2*, whereas retained introns have more motifs for the down-regulated *Hnrnpa1* (Fig 2E). Of note, we could validate the binding of hnRNP A1 to OGT intron 4 and its conservation in human cells using CLIP-seq profiles (Fig S2A), supporting the possibility that the binding events identified may indeed occur in cells. These results position Cpeb4, Elavl2/HuB, and hnRNP A1 as strong candidate RBPs for the regulation of the exon skipping and intron retention events detected in the mouse lungs on the embryonic to postnatal transition. Next, we re-analyzed the AS events that are associated with differentially expressed genes and searched for AS events enriched in Cpeb4, Elavl2/HuB, and hnRNP A1 binding motifs. Of 89 AS events, associated with 59 genes, we identified 21 AS events containing one or more of these binding motifs (Table S10), including genes such as *Abr*, *Aspn*, and *Vegfa*, which has previously been associated with lung development and/or pathology (Yamamoto et al, 2007; Yu et al, 2012; Tian et al, 2019; Ellis et al, 2020). In conclusion, we found a comprehensive and specific AS signature and potential AS regulators involved in the embryonic to postnatal transition.

### Analysis of AS identifies a novel *Vegfa* isoform containing intron 5

Next, to further dissect DAS relevant for the transition between embryonic and postnatal time-points, we focused on *Vegfa* because it has been shown to be essential for lung development (Yamamoto et al, 2007; Ellis et al, 2020) by regulating blood vessel formation (Eichmann & Simons, 2012). Visual inspection of *Vegfa* RNAseq profile revealed an increase in the inclusion of *Vegfa* exon 6 and of intron 5 into processed mRNA transcripts from embryonic to postnatal time-points (Fig 3A). Whereas AS of *Vegfa* exon 6 has been previously described (Bowler & Oltean, 2019), the inclusion of intron 5 in *Vegfa* mature mRNA species has never been reported before. Thus, we characterized in more detail the existence of intron 5-containing isoforms. To identify which *Vegfa* isoform(s) contain(s) intron 5, we analysed RNA extracted from lungs at P5, a stage at which intron 5 retention was evident (Fig 3A). Specifically, from these RNA samples, we produced cDNA and performed end-point PCR amplification of the *Vegfa* intron 5-containing cDNA molecules. For that, we used primer pairs in which one primer anneals with intron 5 and the other anneals with the 5'UTR/exon 1 or with 3'UTR (Fig 3B). The size of the resulting PCR products and the fact that only one band per PCR reaction was obtained (Fig 3C), suggested that the *Vegfa* isoform containing intron 5 also contains all *Vegfa* exons from one to eight. We named this isoform *Vegfa i5*. The sequence of

this newly identified isoform was further confirmed by Sanger sequencing of the amplified PCR products (Fig 3B, bottom panel).

Interestingly, we predicted motif hits for all RBPs in the Catalog of Inferred Sequence Binding Preferences (CISBP-RNA) database along the Vegfa intron 5 sequence and found 19 matches that correspond to differentially expressed genes through development (Table S11). These include the RBPs above identified as candidate regulators of the AS changes, Elavl2/HuB, Cpeb4 and hnRNP A1, suggesting that these may also regulate *Vegfa i5* expression during mouse lung development. These results revealed that a previously unidentified *Vegfa* isoform, *Vegfa i5*, is expressed during lung development.

### *Vegfa* isoforms expression changes during lung development

To further identify which *Vegfa* isoforms, in addition to *Vegfa i5*, are expressed during lung development, we analysed the expression changes of the *Vegfa* isoforms previously annotated in ENSEMBL using Kallisto (Bray et al, 2016). Moreover, we manually annotated the newly identified *Vegfa i5* isoform on Kallisto index. We found that *Vegfa 120*, *164*, *188*, and *i5* are expressed during lung development and that all these isoforms increase in expression from embryonic to postnatal time-points (Fig 4A). This increase is concordant with the increase in total *Vegfa* expression levels in RNAseq during the perinatal period, from E18.5 to P5 (Fig S3A). Remarkably, the isoforms showing the most prominent increase are the ones containing exon 6: *Vegfa 188* and *i5* (absolute fold change from E18.5 to P5 of 7.06 and 4.18, respectively) (Fig 4A). Concordantly, the *Vegfa* Vast DB AS event undergoing changes is the inclusion of exon 6 (Fig S3B).

To characterize the temporal dynamics of AS of *Vegfa* with higher resolution, we collected lungs at E15.5, E17.5, E18.5, E19.5, P0, P5, and P8, as well as two later time-points: P21, at which alveologenesis is still occurring, and adult, at which alveologenesis has already ceased (Fig 4B). The analysis of the expression dynamics of each *Vegfa* isoform at different developmental stages was performed by RT-qPCR. For that, we designed primer pairs that specifically amplify each of the isoforms (Fig 4C). In addition, we designed a pair of primers that amplifies all *Vegfa* isoforms (Fig 4C, primers F2 and R2). The specificity of these primer pairs on the RT-qPCR was validated by the amplification of a single PCR product of the predicted size for each isoform (Fig S3C). This result also excluded the eventual occurrence of spurious amplification of genomic DNA.

In accordance with the RNAseq results, RT-qPCR analysis revealed a progressive increase in total *Vegfa* mRNA levels from E15.5 to the adult stage (Fig S3D), which is associated with an increase in the VEGFA protein levels, as assessed through ELISA analysis of whole lung protein lysates (Fig S3E). From the *Vegfa* isoforms, we found that there is a significant increase in *Vegfa 120* and *164* from E15.5 to P8 (fold change of 4.35 and 2.78, respectively) (Fig 4D). The expression levels of these isoforms remain high in P21 and adult lungs. On the other hand, *Vegfa 188* levels sharply increase at E19.5 (fold change of 18.09 between E15.5 and E19.5), just before birth and then increase even further at P8, P21 and adult (fold change of 28.38, 61.13, and 62.46 between E15.5 and P8, E15.5 and P21, and E15.5 and adult, respectively) (Fig 4D). *Vegfa i5* increases from E15.5 to E17.5 (fold change of 3.89), and further

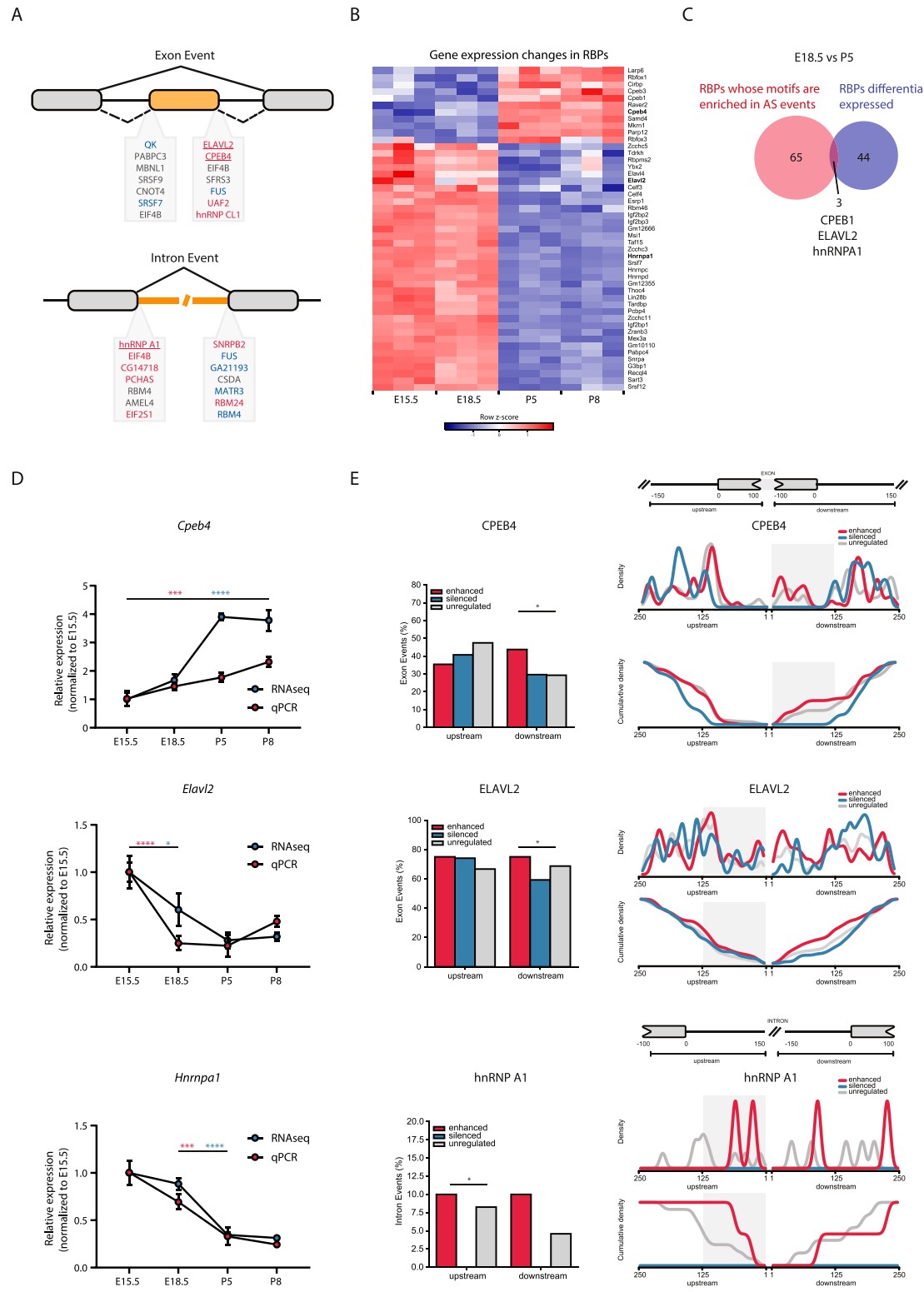

**Figure 2. Candidate RNA-binding proteins (RBPs) for the regulation of differential alternative splicing (DAS) between E18.5 and P5.**
**(A)** RBPs whose motifs are enriched in DAS events occurring in non-differentially expressed genes between E18.5 and P5. Red, enrichment in enhanced versus unregulated. Blue, enrichment in silenced versus unregulated. Gray, depletion in silenced/enhanced versus unregulated. Bold and underlined, RBPs whose motifs are enriched and that undergo differential gene expression between E18.5 and P5. **(B)** Heat map representing gene expression changes of RBPs that undergo differential gene expression between E18.5 and P5 and whose motifs were enriched in the alternative splicing events. Blue and red represent decreased and increased gene expression, respectively, relative to the mean of each gene across the time course (row z-score calculated from $\log_2$ [CPMs + 1]). **(C)** Venn diagram representing the overlap between

increases at P8 (fold change of 14.06 between E15.5 and P8), remaining high at P21 and adult (Fig 4D). Importantly, the increase in *Vegfa 188* and *i5* is much higher than that of *Vegfa 120* and *Vegfa 164* (fold change of 28.38, 14.06 compared to 4.35, 2.78 between E15.5 and P8, respectively). These observations suggest that the relative proportion between *Vegfa* isoforms changes during lung development. Although RT-qPCR allows the accurate quantification of each isoform, it is not the most suitable method to compare relative changes in expression levels between multiple isoforms. To be able to perform this analysis, we performed end-point PCR followed by PAGE. For that, we used a single primer pair that hybridizes at *Vegfa* 5′ and 3′ UTRs and that amplifies the *Vegfa* isoforms *120*, *164* and *188* (Fig 4C, primers F1 and R1). The amplification of these three isoforms was detectable by the presence of three distinct bands with the size corresponding to each of these isoforms (Fig 4E). The absence of additional bands suggests that the *Vegfaxxxb* isoforms are either not expressed during mouse lung development or are expressed at very low levels. Although this pair of primers should theoretically be able to amplify the *Vegfa i5* isoform as well, it does not. This happens probably because of its larger size when compared with the other isoforms (2,511 bp versus 571, 703, and 775 bp), which makes it more difficult to be amplified when in competition with the lower size isoforms. Nevertheless, we could use this technique to evaluate changes in the proportions between the expression of isoforms *Vegfa 120*, *164* and *188*. This analysis revealed that *Vegfa 164* is the isoform more predominantly expressed in bulk lungs at the embryonic time-points, followed by *Vegfa 120*, with only a minor contribution from *Vegfa 188*. From E17.5 onwards, the proportion of *Vegfa 188* gradually increases, reaching a maximum of 70% in the adult. Reciprocally, the relative proportions of both *Vegfa 120* and *Vegfa 164* decrease (Figs 4E and S3F). These results are concordant with previously published results obtained using RNA protection assays from RNA extracted from mouse lungs during development (Ng et al, 2001).

In sum, we found that all the detected *Vegfa* isoforms start to progressively increase before birth and further increase after birth, which coincides with an overall increase in total *Vegfa* levels. However, they increase at distinct rates during lung development: *Vegfa 188* and *Vegfa i5* isoforms undergo a more pronounced differential increase than *Vegfa120* and *Vegfa164*. This demonstrates the occurrence of *Vegfa* AS during lung development towards the expression of the exon 6-containing isoforms. Remarkably, our fine-grained analysis shows an increase in the relative proportion of *Vegfa 188* starting before birth. These observations suggest that *Vegfa* AS coincides with a developmental adaptation to birth.

### *Vegfa* isoforms are differentially expressed between endothelial and epithelial cell populations during lung development

Our results showed that *Vegfa* undergoes AS changes during lung development. However, it was unclear which cell types express which *Vegfa* isoforms and what is their expression dynamics within the different cell types. Previously, *Vegfa* expression was documented in ECs, AT1 and AT2 cells by in situ hybridization (Ng et al, 2001; Compernolle et al, 2002; Greenberg et al, 2002). More recently, genetic LacZ reporters and scRNAseq analyses have reported the expression of *Vegfa* only in AT1 and ECs (Treutlein et al, 2014; Yang et al, 2016; Gillich et al, 2020; Ellis et al, 2020). Yet, these studies have only examined global levels of *Vegfa* transcripts. To unravel which cells express the different *Vegfa* isoforms, we isolated various lung cell types at different time-points of development and assessed how *Vegfa* isoforms expression varies within each cell type (Fig 5A). We examined lungs at E15.5, E17.5, E18.5, E19.5, P0, P5, P8, P21, and adult, in accordance with our analysis in bulk lungs. To isolate the different lung cell types, we dissociated the lung tissue into a single-cell suspension and performed FACS using antibodies for cell type–specific markers. The combination of these markers allowed the isolation of cell populations enriched for ECs (CD31 single positive [SP]: CD31$^+$, EpCAM$^-$, and CD45$^-$), epithelial cells (EpCAM SP: CD31$^-$, EpCAM$^+$, and CD45$^-$), immune cells (CD45 SP: CD31$^-$, EpCAM$^-$, and CD45$^+$) and mesenchymal cells, such as alveolar myofibroblasts and pericytes (triple negative [TN]: CD31$^-$, EpCAM$^-$, and CD45$^-$) (Fig S4A). The analysis of gene expression changes was performed by RNA extraction from the isolated cell populations followed by RT-qPCR or end-point PCR (Fig 5A).

We first evaluated the quality of the isolation of the different cell populations. For that, we examined the expression of cell type–specific markers in the different cell populations collected at P5 by RT-qPCR. We analysed the mRNA levels of *Pecam1* (CD31) and *Cdh5* as pan-endothelial-specific markers, of *Prox1* as a marker for lymphatic ECs, of *Epcam* and *Cdh1* as pan-epithelial–specific markers, and of *Cd45* as a pan-immune cell marker. We identified that only CD31 SP population expresses *Pecam1* (CD31) and *Cdh5* (Fig S4B). However, this population also expresses high levels of *Prox1* (Fig S4B), suggesting that CD31 SP contains a mixture of blood and lymphatic ECs. We also showed that only EpCAM SP population expresses *Epcam* and *Cdh1*, and that only CD45 SP expresses *Cd45*, whereas TN cells do not express any of these markers (Fig S4B). In addition, by analyzing the mRNA levels of *Aqp5*, *Sftpc* and *Foxj1*, markers of epithelial alveolar AT1, AT2, and ciliated cells, respectively, we demonstrated that the EpCAM SP population is composed by a mixture of these lung epithelial subtypes (Fig S4B).

To estimate the contribution of each endothelial and epithelial subtypes in CD31 SP and EpCAM SP populations, we analysed the expression of cell type–specific markers at the protein level in single cell suspensions before sorting (pre-sort) and after sorting using cytospin preparation (Koh, 2013) and immunostaining. We observed that CD31 SP population is highly enriched for CD31-positive cells, as compared with the pre-sort sample. Of these, about 60% are also positive for ERG, a marker of blood ECs (Fig S4C). The remaining fraction of CD31-positive cells is likely composed of

---

the RBPs with enriched motifs in DAS events occurring in non-differentially expressed genes between E18.5 and P5 and those that undergo differential gene expression genes between E18.5 and P5. **(D)** Expression changes of *Cpeb4*, *Elavl2*, and *Hnrnpa1* on bulk lungs at different developmental time-points analysed by RNAseq (blue) and RT-qPCR (red). N = 3 for each time-point. *P*-value from one-way ANOVA with Tukey correction for multiple testing. **(E)** Left: Bar plots show the relative number of events with at least one motif hit for each RBP normalized by the total number of respective events. Right: Density plots and cumulative density plots show the positional distribution of each RBP binding motif in regulated events against a background of unregulated events. These show where the motifs are more concentrated, and thus where the enrichment test was more significant (shaded area). Statistical significance was calculated using Matt *test_regexp_enrich* (*P* < 0.01).

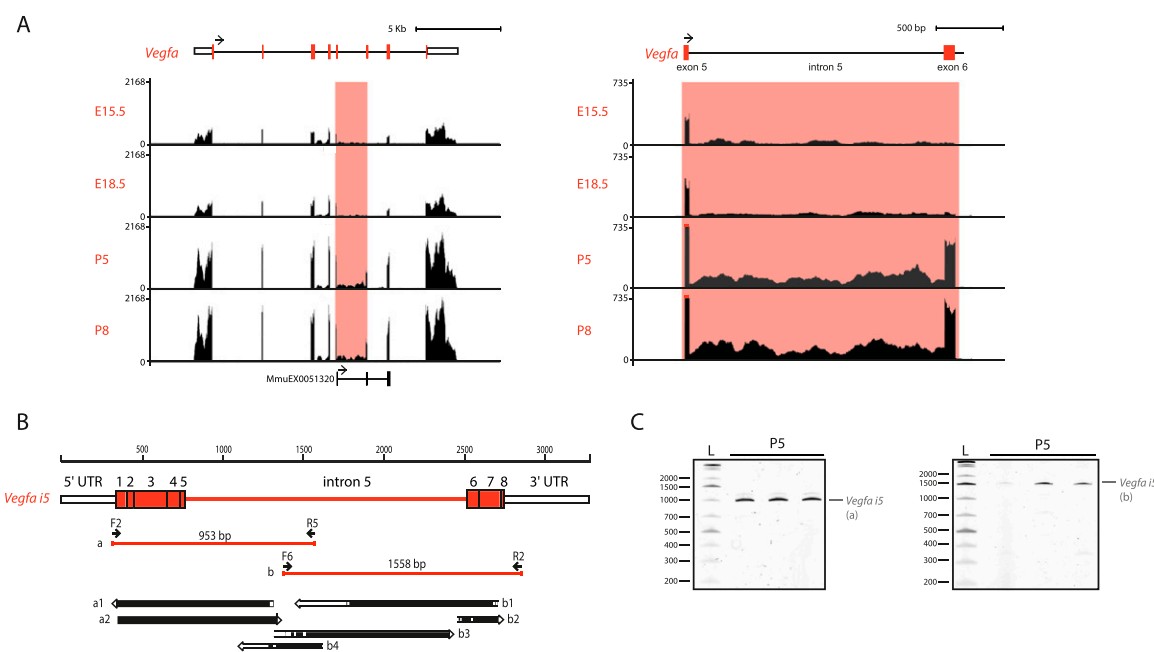

**Figure 3. Identification of a novel *Vegfa* isoform – Vegfa *i5*.**
**(A)** Left: RNAseq profile of *Vegfa* from bulk lung at different developmental time-points. VAST-TOOLS tracks representing previously annotated alternative splicing events associated with *Vegfa*. Right: Higher magnification of the region containing intron 5 and exon 6. **(B)** Schematic representation of the identified *Vegfa i5* isoform. Primer pairs used and size of the respective end-point PCR amplicons obtained is indicated. The alignment of sequenced PCR products with the predicted *Vegfa i5* isoform is represented on the bottom panel. Black bars represent full alignment of the sequence, white bars indicate no alignment. Poor sequencing quality at the ends of the sequenced fragments justifies the predominance of non-aligned sequences in these regions. **(C)** TBE-urea PAGE gels of the *Vegfa i5* amplification products indicated in Fig 3B.

lymphatic ECs, as suggested by the enrichment in expression of the lymphatic ECs marker *Prox1* in this population (Fig S4B). EpCAM SP population is enriched for AQP5-positive and SFTPC-positive cells (21.2% and 79.2% on average, respectively), demonstrating that this population is enriched for both epithelial AT1 and AT2 cells (Fig S4D and E). Altogether, these results validate the quality of our method for isolation of different lung cell types.

We then analysed the expression changes of each *Vegfa* isoform in each cell population along lung development by RT-qPCR. Total *Vegfa* expression is highest in EpCAM SP at all time-points analysed, followed by CD31 SP population (Fig S5A). In addition, we found that CD31 SP and EpCAM SP populations reveal the highest fold changes in total *Vegfa* expression during development, when compared with CD45 SP and TN populations (fold change between E15.5 and adult of 45.21, 22.90, 4.70, and 3.87, respectively) (Fig 5B). Therefore, to characterize *Vegfa* AS, we focused our analysis on EpCAM SP and CD31 SP populations. We found that, in the CD31 SP population, *Vegfa 120*, *164*, *188* and *i5* steadily increase during lung development from E18.5 to adult (Fig 5C). The analysis of *Vegfa* isoforms relative proportions by end-point PCR and PAGE revealed that *Vegfa 164* is the most abundant isoform and that there was no significant change in the proportion between the *Vegfa* isoforms during lung development in CD31 SP population (Figs 5E and F and S5B). These results suggest that AS of *Vegfa* does not significantly change in the EC population.

In EpCAM SP population, *Vegfa 120* and *Vegfa164* increase before birth from E15.5 onwards, peaking at P0, after which decrease until

P21. *Vegfa 188* and *Vegfa i5* increase from E17.5 until P5 (Fig 5D). In the adult, *Vegfa 120*, *Vegfa 164*, and *Vegfa 188* exhibit the highest levels both in EpCAM SP and CD31 SP populations (Fig 5C and D).

Analysis of the relative proportions of *Vegfa* isoforms in EpCAM SP population through end-point PCR and PAGE showed that there is a prominent increase in the relative proportion of *Vegfa 188* in epithelial cells throughout lung development (Figs 5E and F and S5C). Whereas at embryonic time-points *Vegfa 164* and *Vegfa 120* are the predominant isoforms, the proportion of *Vegfa 188* increases progressively throughout development and, at postnatal time-points, it becomes the most abundant isoform expressed in EpCAM SP cells (Figs 5E and F and S5C). These results suggest that the signature of AS for *Vegfa* that we have detected on bulk lungs is associated with changes occurring specifically in the epithelial lineage.

In sum, our results suggest that the endothelial and epithelial lineages express the highest *Vegfa* levels and that *Vegfa* expression increases in both during lung development. Whereas in ECs, *Vegfa 164* is always the predominant isoform, in epithelial cells, there is a marked increase in the proportion of *Vegfa 188*, from E17.5 onwards. These results suggest a cell type–specific AS of *Vegfa* in epithelial cells during lung development at the perinatal period.

We performed expression analysis on sorted cell populations of the RBPs we have identified as candidate regulators of AS on the developing lung, *Cpeb4*, *Elavl2*, and *Hnrnpa1* (Fig 2). We found that *Cpeb4* increases in expression during lung development on CD31 SP,

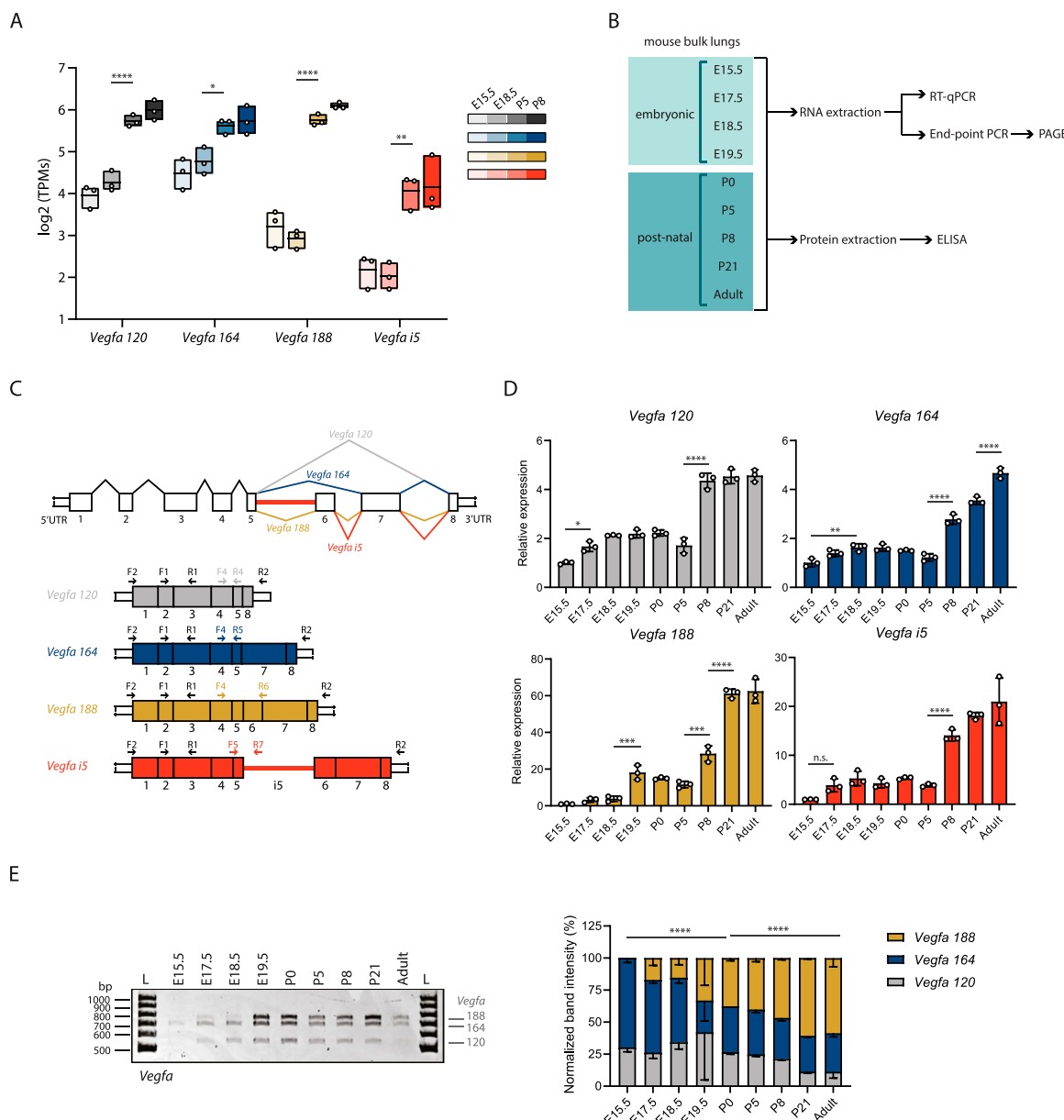

**Figure 4. *Vegfa* isoforms are dynamically expressed during lung development.**
**(A)** Quantification of expression of *Vegfa* alternative splicing isoforms at different developmental time-points using Kallisto. TPM, transcripts per million. *P*-value from one-way ANOVA with Tukey correction for multiple testing. Data represented as mean ± max/min. **(B)** Schematic representation of workflow: Bulk lungs at different developmental time-points were collected. RNA or protein was extracted. cDNA was produced from RNA. Expression analysis was performed by RT-qPCR. Analysis of the relative proportions between the expression of different isoforms was performed by end-point PCR and PAGE. Protein levels were quantified by ELISA. **(C)** Schematic representation of the *Vegfa* isoforms analysed in this study. Each exon is represented by a number and *i5* represents intron 5. Primer pairs used to amplify all *Vegfa* isoforms and each isoform individually in RT-qPCR or end-point PCR are indicated in figure. **(D)** Expression changes of *Vegfa* isoforms from bulk lungs at different developmental time-points analysed by RT-qPCR. N = 3 for each time-point. *P*-value from one-way ANOVA with Tukey correction for multiple testing. **(E)** Left: Representative TBE-urea PAGE from PCR products obtained from cDNA samples from bulk lungs at different developmental time-points. Fragments were amplified using primer pair F2-R2. Bands represent *Vegfa 188*, *164*, and *120* isoforms. Right: Quantification of the normalized relative proportions between the expression levels of *Vegfa* isoforms on bulk lungs at different developmental time-points. N = 3 for each time-point. *P*-value from chi-square test. TBE-urea PAGE gels used for this quantification is represented in Fig S3F.

EpCAM SP, and TN cell populations, whereas *Elavl2* and *Hnrnpa1* decrease in expression during lung development in EpCAM SP and TN cell populations (Fig S5D). Remarkably, their highest expression fold changes within the developmental time interval analysed occur in EpCAM SP population from E15.5 to E18.5 (Fig S5D), supporting the hypothesis that the expression changes of these RBPs in EpCAM SP cell population may drive AS changes, including those observed for *Vegfa* in this same cell type (Fig 5D–F).

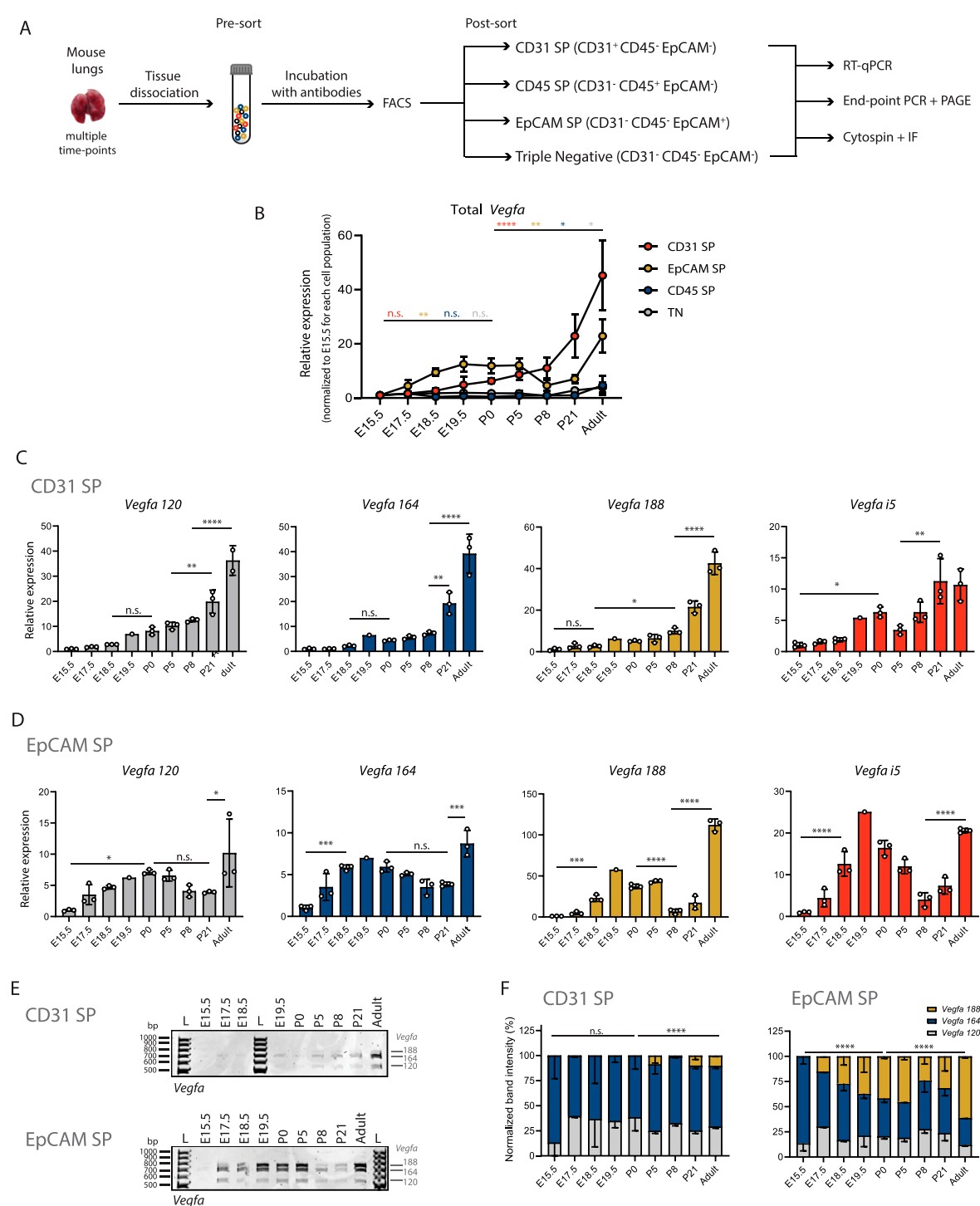

**Figure 5.  *Vegfa* isoforms are differentially expressed between CD31 SP and EpCAM SP populations during lung development.**
**(A)** Schematic representation of workflow: Cell suspensions from lungs at different developmental time-points were collected (pre-sort sample). Several lung cell populations were isolated by FACS. Samples were analysed for the expression of indicated surface markers (CD45, CD31, and EpCAM). CD31 SP populations are enriched for endothelial cells, CD45 SP populations are enriched for immune cells, EpCAM SP populations are enriched for epithelial cells, triple negative (TN) are enriched for mesenchymal cells. RNA was extracted from isolated cell populations and cDNA produced. Expression analysis was performed by RT-qPCR. Analysis of the relative proportions between the expression of different isoforms was performed by end-point PCR and PAGE. Alternatively, isolated cell populations were processed by cytospin and immunofluorescence (IF) was performed. **(B)** Expression changes of total *Vegfa* from sorted cell populations at different developmental time-points analysed by RT-qPCR. Expression values normalized to E15.5 for each cell population. N = 3 for each time-point. *P*-value from one-way ANOVA with Tukey correction for multiple testing. **(C)** Expression changes of *Vegfa* isoforms from CD31 SP cell population at different developmental time-points analysed by RT-qPCR. N = 3 for each time-point except for E19.5. N = 1 for E19.5. *P*-value from one-way ANOVA with Tukey correction for multiple testing. **(D)** Expression changes of *Vegfa* isoforms from EpCAM SP cell population at different developmental time-points analysed by RT-qPCR. N = 3 for each time-point except for E19.5. N = 1 for E19.5. *P*-value from one-way ANOVA with Tukey correction

### *Vegfa* undergoes AS in epithelial AT1 cells during lung development

The EpCAM SP population is composed of multiple subtypes, such as epithelial alveolar AT1, AT2, and epithelial bronchiolar ciliated cells (Fig S4A and C–E). To dissect the contribution of each epithelial cell subtype for the expression of *Vegfa* isoforms, we further subdivided EpCAM-positive population into AT1- and AT2-enriched subpopulations by FACS in P5 lungs (Fig 6A), a stage at which *Vegfa* expression in the epithelial lineage is high (Fig S5A). For that, we used the marker major histocompatibility complex class II (MHC II) which, in combination with EpCAM, has been previously shown to discriminate between AT1 (EpCAM$^{low}$ MHC II$^-$), AT2 (EpCAM$^{high}$ MHC II$^+$), and ciliated cells (EpCAM$^{high}$ MHC II$^-$) (Hasegawa et al, 2017). We could identify within the EpCAM-positive cell population the presence EpCAM$^{low}$ MHC II$^-$ and EpCAM$^{high}$ MHC II$^+$ subpopulations putatively corresponding to AT1 and AT2, respectively (Fig S6A). We detected very few EpCAM$^{high}$ MHC II$^-$ cells, putative ciliated cells (Fig S6A), and therefore this subpopulation was not further analysed.

mRNA analysis of the selected subpopulations by RT-qPCR revealed that none of these two subpopulations expresses *Pecam1* (CD31) nor *Cd45*, whereas both express *Epcam* mRNA (Fig S6B). In addition, by RT-qPCR, we found that EpCAM$^{low}$ MHC II$^-$ cells express high levels of *Aqp5* mRNA and low levels of *Sftpc* mRNA, whereas EpCAM$^{high}$ MHC II$^+$ cells express high levels of *Sftpc* mRNA and low levels of *Aqp5* mRNA (Fig S6C), suggesting that indeed they are enriched for AT1 and AT2 cells, respectively. None of these two subpopulations is enriched for a marker of epithelial bronchial ciliated cells, *Foxj1*, as compared with EpCAM SP population (Fig S6C), suggesting that this cell type is not present on these two sorted subpopulations. Complementarily, by cytospin and immunofluorescence, we found that 54.5% of EpCAM$^{low}$ MHC II$^-$ cells are AQP5-positive cells, whereas 87.8% of EpCAM$^{high}$ MHC II$^+$ cells are SFTPC-positive cells (Fig S6D and E). Thus, this analysis revealed that this FACS gating strategy enables the isolation of subpopulations enriched for epithelial AT1 and AT2 cells.

Analysis of *Vegfa* expression levels in AT1- and AT2-enriched subpopulations through RT-qPCR revealed that *Vegfa* isoforms expression is markedly higher in AT1 cells than in AT2 cells (Fig 6B), in agreement with published scRNAseq results (Yang et al, 2016; Raredon et al, 2019; Ellis et al, 2020). Whereas there is only a moderate increase in the expression of *Vegfa 164*, the levels of expression of *Vegfa 188* and *Vegfa i5* are markedly higher in AT1 than in AT2 (fold enrichment of 2.87, 31.60, and 14.53, respectively) (Fig 6B). The levels of *Vegfa 120* are not significantly distinct between both subpopulations (Fig 6B). Through end-point PCR and PAGE, we found that *Vegfa 188* is the isoform whose expression is predominant in AT1 cells at P5 (Figs 6C and S6F). Our results suggest that the expression of *Vegfa 188* in EpCAM SP population is due to its expression in AT1 cells.

*Vegfa 188* starts to be expressed in EpCAM SP population at E17.5 (Fig 5D). Interestingly, it is just before this time-point that AT1 cells start differentiating from AT1/AT2 progenitor cells (Yang et al, 2016; Wang et al, 2018; Vila Ellis & Chen, 2021). This led us to question if the increase in *Vegfa 188* proportion in EpCAM SP population is solely due to an increase in the fraction of AT1 within EpCAM SP population during development, as we detected by FACS (Fig S6G), or if the predominance of *Vegfa 188* in AT1 cells also increases during development. To disentangle between these two possibilities, we analysed *Vegfa* isoforms expression in AT1 cells at E18.5, a time-point just after their specification, and compared it to that in AT1 cells at P5. By RT-qPCR analysis, we found that *Vegfa* isoforms expression increases in AT1 cells from E18.5 to P5. This increase is mild for *Vegfa 120* and *164*, but it is higher for *Vegfa 188* and *Vegfa i5* (fold change of 1.37, 1.38, 2.54, and 2.08, respectively) (Fig 6D). Accordingly, although at E18.5 *Vegfa 188* proportion is already 42% of the total, its proportion further increases at P5 to 49% (Figs 6E and S6F). Altogether, our results suggest that *Vegfa* undergoes AS changes in epithelial AT1 cells during lung development.

### VEGFA 188 specifically drives the formation of a specialized type of alveolar capillary ECs—Car4-positive ECs

AT1-derived *Vegfa* expression is important for the specification of Car4-positive ECs (also known as aCap or aerocytes), a specialized alveolar EC subtype (Gillich et al, 2020; Ellis et al, 2020). Remarkably, the specification of aerocytes occurs just before birth, at around E19.5 (Ellis et al, 2020). It is, however, unclear if aerocyte differentiation depends on overall levels of *Vegfa* or if a specific *Vegfa* isoform drives this effect. In this study, we identified a significant increase in *Vegfa 188* expression from E17.5 to E19.5 in AT1 cells, reaching around 50% of total *Vegfa* transcripts. This observation led us to hypothesize that Car4-positive aerocyte differentiation may be controlled by *Vegfa 188* specifically in the developing lung alveoli. To test this, we cultured E17.5 mouse lung explants on an air-liquid interface (Prince et al, 2004). Explants were cultured in medium supplemented with either control, VEGFA 164 or VEGFA 188 for 24 h (Fig 7A). Immunofluorescence analysis revealed a significant increase in the fraction of Car4-positive aerocytes on explants cultured in medium supplemented with VEGFA 188, as compared with those cultured in that containing VEGFA 164 or on control conditions (Fig 7B–D). These results constitute the first evidence that aerocyte differentiation is specifically controlled by the VEGFA 188 isoform during lung development.

## Discussion

Lung alveolar formation starts during late embryonic development and continues postnatally during the first weeks after birth. The distinct cell types that compose the developing alveoli experience dramatic changes at birth. Oxygen tension, mechanical stretch due to respiratory movements, and blood flow rates dramatically

---

for multiple testing. **(E)** Representative TBE-urea PAGE from PCR products obtained from cDNA samples from CD31 SP and EpCAM SP cell population at different developmental time-points. **(F)** Quantification of the normalized relative proportions between the expression levels of *Vegfa* isoforms on CD31 SP and EpCAM SP cell population at different developmental time-points. N = 3 for each time-point. *P*-value from chi-square test. TBE-urea PAGE gels used for this quantification are represented in Fig S5B and C.

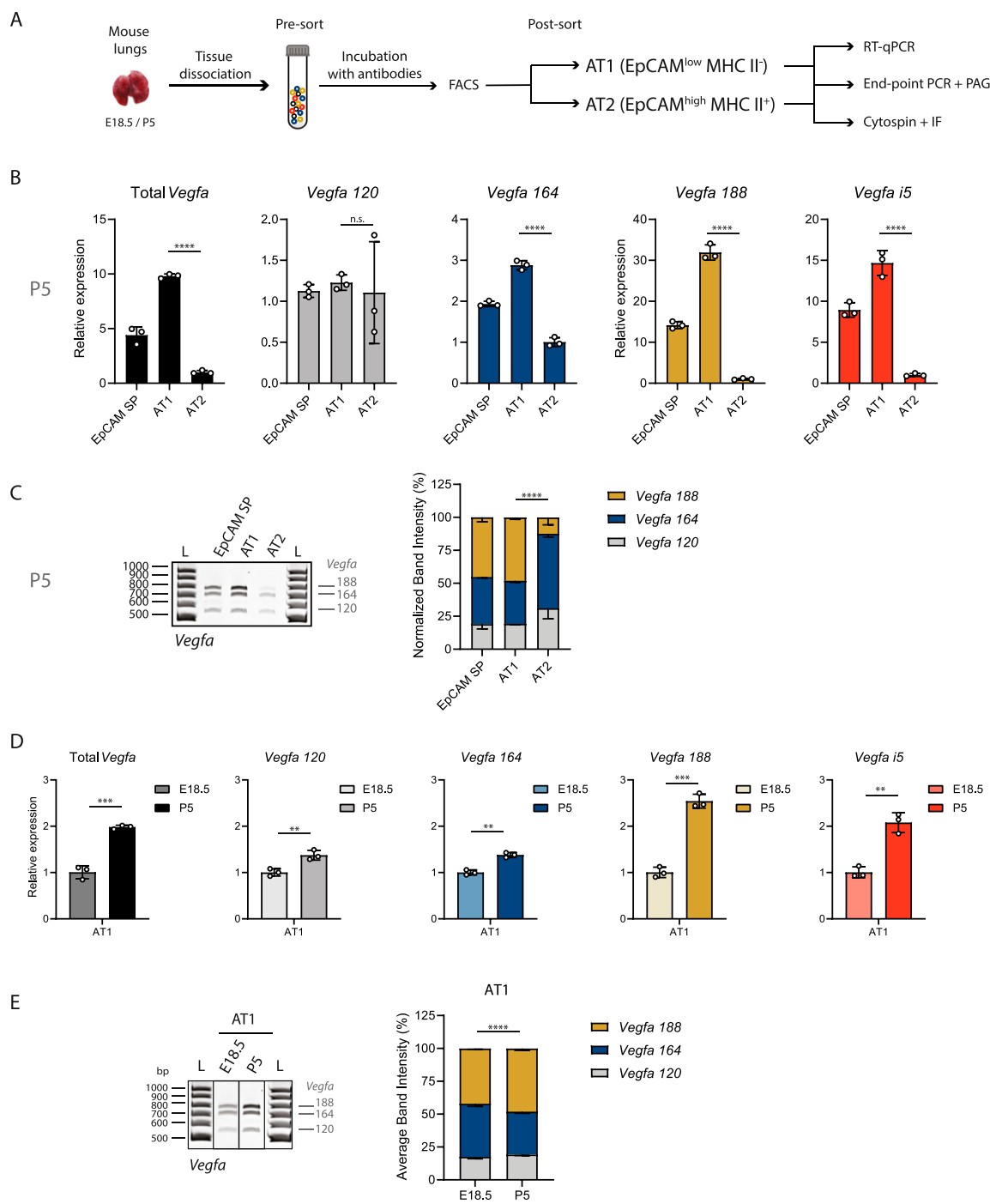

**Figure 6. *Vegfa* undergoes alternative splicing towards *Vegfa 188* in epithelial AT1 during lung development.**
**(A)** Schematic representation of workflow: Cell suspensions from lungs at E18.5 and P5 were collected (pre-sort samples). Several lung cell populations were isolated by FACS. Samples were analysed for the expression of indicated surface markers (EpCAM and MHC II). EpCAM$^{low}$ MHC II$^-$ populations are enriched for AT1 cells, EpCAM$^{high}$ MHC II$^+$ populations are enriched for AT2 cells. RNA was extracted from isolated cell populations and cDNA produced. Expression analysis was performed by RT-qPCR. Analysis of the relative proportions between the expression of different isoforms was performed by end-point PCR and PAGE. Alternatively, isolated cell populations were processed by cytospin and immunofluorescence (IF) was performed. **(B)** Expression changes of *Vegfa* isoforms from EpCAM SP, AT1 and AT2 cell population at P5 were analysed by RT-qPCR. N = 3 for each time-point. *P*-value from unpaired *t* test. **(C)** Left: Representative TBE-urea PAGE from PCR products obtained from cDNA samples from EpCAM SP, AT1, and AT2 cell populations at P5. Right: Quantification of the normalized relative proportions between the expression levels of *Vegfa* isoforms on EpCAM SP, AT1, and AT2 cell populations at P5. N = 3 for each time-point. *P*-value from chi-square test. TBE-urea PAGE gels used for this quantification are represented in Figs S5C and S6F. **(D)** Expression changes of *Vegfa* isoforms from AT1 cell population at E18.5 and P5 were analysed by RT-qPCR. N = 3 for each time-point. *P*-value from unpaired *t* test. **(E)** Left: Representative TBE-urea PAGE from PCR products obtained from cDNA samples from AT1 cell populations at E18.5 and P5. Right: Quantification of the normalized relative proportions between the expression levels of *Vegfa* isoforms on AT1 cell populations at E18.5 and P5. N = 3 for each time-point. *P*-value from chi-square test. TBE-urea PAGE gels used for this quantification are represented in Fig S6F.

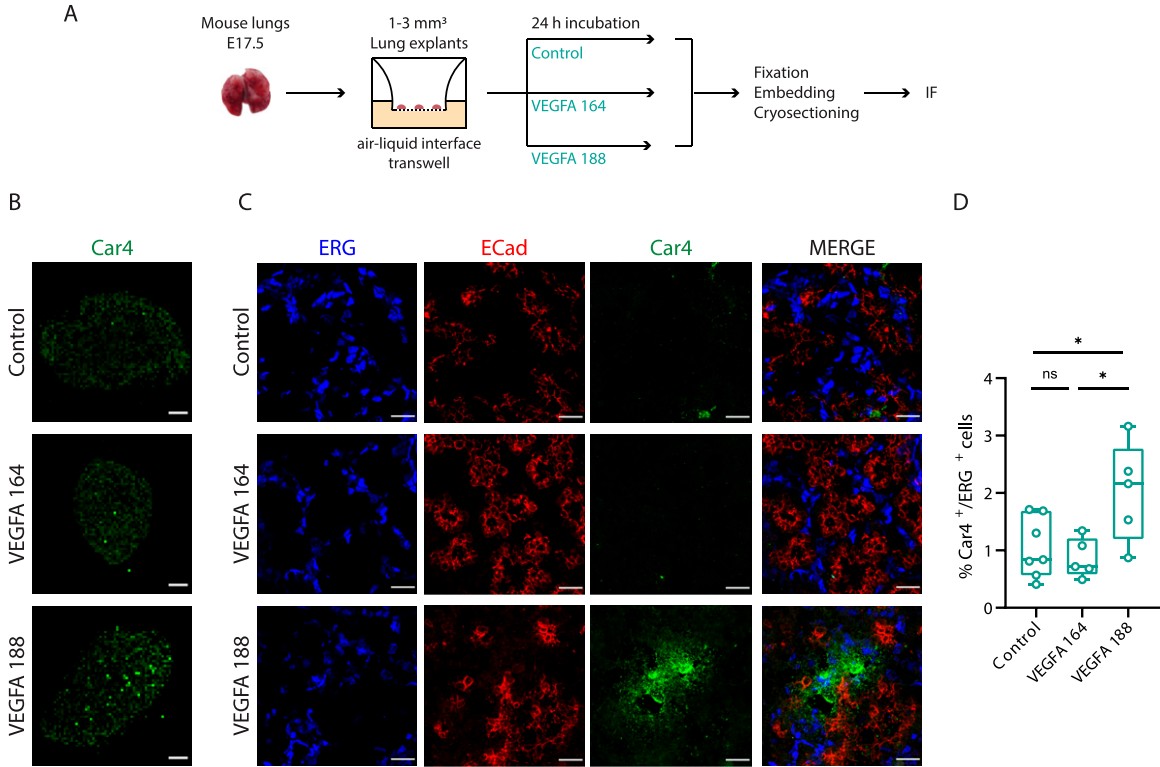

**Figure 7. Vegfa 188 selectively induces the specification of Car4-positive endothelial cells.**
**(A)** Schematic representation of workflow: E17.5 lung explants were isolated and cultured on a Transwell at an air-liquid interface. Control and mouse recombinant VEGFA 164 or VEGFA 188 were added to the medium in contact with the explants. After 24 h, the explants were collected and processed for immunofluorescence (IF).
**(B)** Low-magnification images of immunofluorescence of Car4 (green) in E17.5 mouse explants treated with control, VEGFA 164, or VEGFA 188 for 24 h. Scale bar, 200 $\mu$m.
**(C)** High magnification images of immunofluorescence of ERG (gray), Ecad (red), and Car4 (green) in E17.5 mouse explants treated with control, VEGFA 164, or VEGFA 188 for 24 h. Scale bar, 20 $\mu$m. **(D)** Quantification of the percentage of Car4-positive/ERG-positive cells in E17.5 mouse explants treated with control, VEGFA 164 or VEGFA 188 for 24 h. Data from five to seven independent biological replicates for each condition with a total of at least 10,000 cells quantified per condition. Data are shown as mean ± SD. $P$-value from one-way ANOVA with Tukey correction for multiple testing.

increase. However, how the distinct lung cell types adapt to such dramatic changes is still not fully understood.

AS is a wide phenomenon driving developmental changes. However, the regulation of AS during lung development at a genome-wide level had not been explored before. This work represents the first comprehensive analysis of the precise dynamics of AS during lung development at a genome-wide scale. Our analysis uncovered the occurrence of AS in the mouse lung during the embryonic-to-postnatal transition. We identified numerous genes that undergo AS at this transition, creating two clusters that exhibit distinct kinetics between embryonic and postnatal stages. We identified AS changes in key cell–cell adhesion complexes and signaling pathways known to regulate intercellular communication between epithelial, endothelial and other lung cell types during lung development: adherens and tight junctions, and VEGFA and Hippo signaling pathways (Yamamoto et al, 2007; Kato et al, 2018; Nantie et al, 2018; Ellis et al, 2020). Overall, this raises the hypothesis that regulation of AS can be a means of modulating intercellular communication in lungs towards functional respiration. Remarkably, the number of AS changes in genes that do not undergo gene expression changes was considerably higher than in genes that undergo gene expression changes. Thus, it is tempting to speculate that the program of adaptation to birth through AS is largely independent of the gene expression program operating at this stage. Therefore, we postulate that further investigation into the mechanisms governing AS in lung could be relevant for development and pathology. Supporting this vision, several studies have already associated AS and different lung conditions, such as lung cancer, idiopathic pulmonary fibrosis, and chronic obstructive pulmonary disease (Kusko et al, 2016; Coomer et al, 2019).

Through bioinformatics analysis, we identified Cpeb4, Elavl2, and hnRNP A1 as RPBs potentially regulating DAS in mouse lungs. Cpeb4 regulates polyadenylation of the 3′UTR of mRNA transcripts and thus, the stability and translational output and has previously been shown to bind to and regulate *Vegfa* (Calderone et al, 2016). hnRNP A1 has been shown to bind to intronic or exonic splice silencers to regulate splicing of alternative exons (Baralle & Giudice, 2017). Elavl2/HuB belongs to the family of Hu proteins that have been implicated in the regulation of AS and alternative polyadenylation (Pascale et al, 2008). *Cpeb4* or *Elavl2* knockout mice are viable (Liu et al, 2017; Kato et al, 2019), thus suggesting a mild effect in lung development, whereas *Hnrnpa1* knockout mice die perinatally with cardiac defects but analysis of lungs was not reported (Liu et al, 2017). Thus, further investigation on the biological relevance of each of these factors, and combinatorial effects, is warranted to validate their roles in lung development.

**Key resources table.**

| Reagent or resource | Source | Identifier |
|---|---|---|
| Antibodies | | |
| Rat anti-Mouse APC CD31 | BD Pharmingen | 551262 |
| Mouse anti-Mouse FITC CD45 | Covance | 109806 |
| Rat anti-Mouse PE CD326 (Ep-CAM) | Covance | 118206 |
| Rat anti-Mouse PerCP/Cy5.5 MHC II | BioLegend | 107626 |
| Pro-SFPTC, Rabbit | SevenHills BioReagents | WRAB-9337 |
| CD31 (PECAM1), Goat | R&D | AF3628 |
| Aquaporin 5, Rabbit | Merck | 178615 |
| ERG, Rabbit | Abcam | ab92513 |
| E-cadherin (ECCD-2), Rat | Invitrogen | 13-1900 |
| Carbonic Anhydrase IV/CA4 (Car4), Goat | R&D | AF2414 |
| Donkey anti-Rabbit Alexa 568 | Thermo Fisher Scientific | A10042 |
| Donkey anti-Goat Alexa 647 | Thermo Fisher Scientific | A21447 |
| Donkey anti-Goat Alexa 555 | Thermo Fisher Scientific | A21432 |
| Donkey anti-Rabbit Alexa 488 | Thermo Fisher Scientific | A21206 |
| Chemicals, and recombinant proteins | | |
| PBS | Sigma-Aldrich | P4417 |
| HBSS | Gibco | 14175-053 |
| DMEM | Gibco | 11320-074 |
| DMEM/F-12, no phenol red | Gibco | 21041-025 |
| L-Ascorbic acid | Merck | A4544-25G |
| Penicillin–streptomycin (10,000 U/ml) | Gibco | 15140-122 |
| Mouse VEGFA 164 | ReliaTech | M30-004 |
| Mouse VEGFA 188 | ReliaTech | M30-095 |
| Albumin bovine fraction V (BSA) | Nzytech | MB04602 |
| RNase-OFF | Enzifarma | 9037 |
| Liberase Research Grade | Roche | 5401119001 |
| DNase I, grade II, from bovine pancreas | Roche | 10104159001 |
| Ethylenediaminetetra–acetic acid (EDTA) | VWR | VWRC20301.290 |
| RBC lysis buffer 10× | BioLegend | 420301 |
| RNase-free water | Sigma-Aldrich | 3098 |
| Trypan Blue | Sigma-Aldrich | T8154 |
| Live/Dead Fixable Near-IR Dead Cell Stain Kit | Invitrogen | LTI L10119 |
| Hepes | Sigma-Aldrich | H3375-250G |
| TRIzol Reagent | Alfagene/Ambion | 15596026 |
| Chloroform | Merck Millipore | 1.02445.1000 |
| Glycogen | Sigma-Aldrich | G1767-1VL |
| 3M Sodium acetate | VWR | 27653,26 |
| Ethanol absolute | VWR | 20821,33 |
| Phenol-chloroform-isoamyl alcohol (25:24:1) | Sigma-Aldrich | P2069 |
| Isopropanol | VWR | 1.09634.1000 |
| Power SYBR Green PCR Master Mix | Thermo Fisher Scientific | 4368706 |
| Tris base | Nzytech | MB01601 |

**Continued**

| Reagent or resource | Source | Identifier |
|---|---|---|
| Agarose | Nzytech | MB14402 |
| GreenSafe Premium | Nzytech | MB13201 |
| DNA Gel Loading Dye (6X) | Thermo Fisher Scientific | R0611 |
| GeneRuler 1 Kb Plus DNA ladder | Thermo Fisher Scientific | SM1333 |
| GeneRuler 100 bp DNA ladder | Thermo Fisher Scientific | SM0241 |
| Urea | Sigma-Aldrich | U5378 |
| Ammonium persulfate | Sigma-Aldrich | A3678-25G |
| Ficoll 400 | Enzymatic | BP525-25 |
| Sodium chloride | VWR | 27810,295 |
| Triton X-100 | Sigma-Aldrich | T8787 |
| DTT | Enzo Life Sciences | ALX 280 001 G010 |
| Halt protease and phosphatase inhibitor cocktail | Thermo Fisher Scientific | 78446 |
| Acrylamide/bis-Acrylamide (29:1 solution) | Nzytech | MB04501 |
| Hydrochloric acid | VWR | 20252,335 |
| N,N,N′,N′-TETRAMETHYLETHYLENEDIAMINE | Sigma-Aldrich | T9281-25ML |
| Bromophenol blue | Merck | 108122 |
| RNeasy plus mini kit | QIAGEN | 50974136 |
| RNeasy plus micro kit | QIAGEN | 50974034 |
| DNase I recombinant, RNase-free | Roche | 4716728001 |
| High-capacity RNA-to-cDNA Kit | Applied Biosystems | LTAB 4387406 |
| Q5 Hot Start High-Fidelity DNA polymerase | BioLabs | M0493S |
| GoTaq G2 flexi DNA polymerase | Promega | PROMM7805 |
| Wizard SV gel and PCR clean-up system | Promega | A9281 |
| Pierce BCA protein assay kit | Thermo Fisher Scientific | 23227 |
| Mouse VEGFA Quantikine ELISA kit | R&D | MMV00 |
| Paraformaldehyde 16% solution | Electron Microscopy Sciences | 15710 |
| Sucrose | Sigma-Aldrich | S7903-1KG |
| OCT compound mounting medium | VWR | VWRC361603E |
| Fetal bovine serum | Gibco | 10500-064 |
| Experimental model organisms | | |
| C57Bl/6J mice (*Mus musculus* Linnaeus) | Charles River Laboratories | N/A |
| Oligonucleotides | | |
| Primers used in PCR are listed in the methods section | Sigma-Aldrich and Eurofins | This manuscript |
| Bioinformatics tools and software | | |
| Kallisto | | https://github.com/pachterlab/kallisto |
| VastDB | | https://vastdb.crg.eu/wiki/Main_Page |
| VAST-TOOLS | | https://github.com/vastgroup/vast-tools |
| Matt | | http://matt.crg.eu/ |
| DAVID | | https://david.ncifcrf.gov/tools.jsp |
| QuantStudio real-time PCR software | Applied Biosystems | 4486400 |
| Prism 8 Software | GraphPad | https://www.graphpad.com/scientific-software/prism/ |
| Image Lab Software | Bio-Rad | 1709690 |
| SnapGene Software | GSL Biotech | https://www.snapgene.com |

**Continued**

| Reagent or resource | Source | Identifier |
|---|---|---|
| FLOW JO Software | BD Life Sciences | https://www.flowjo.com/ |
| ZEN software (black edition) | Carl Zeiss | N/A |
| Fiji/ImageJ Software | NIH | https://imagej.nih.gov/ |
| Microsoft Excel 2016 | Office | N/A |
| Others | | |
| Sanger sequencing | Eurofins\|genomics/GATC services | N/A |

We have further explored AS of *Vegfa* gene. Although changes in the relative proportion of *Vegfa* isoforms have previously been described to occur during lung development (Healy et al, 2000; Ng et al, 2001; Greenberg et al, 2002), it was not known which cell types in the lung express the different *Vegfa* isoforms, nor if the switch in *Vegfa* AS occurs in a tissue- or cell type–specific manner. We identified that *Vegfa* transcript levels increase both in ECs and in AT1 cells during alveolar development. However, whereas in the embryonic period, *Vegfa 164* is the predominant isoform expressed in both cell populations, in the postnatal period, *Vegfa 188* becomes predominant specifically in AT1 cells. Our results suggest the occurrence of a cell type–specific AS of *Vegfa* in AT1 cells during late lung development. Finally, we have shown that VEGFA 188, and not VEGFA 164, drives the specification of Car4-positive alveolar ECs. This result suggests the existence of a developmental program driving AS and the sharp increase in *Vegfa 188* isoform expression from E18.5 to E19.5 in AT1 cells. The alteration in *Vegfa* AS pattern results in a change in the intercellular communication between epithelial AT1 and EC types to promote organotypic endothelial cell specification in the developing mouse alveoli. This mechanism may constitute a pre-birth adaptation of lungs to the postnatal life.

We also found a marked increase in *Vegfa* expression during lung development in ECs, in which *Vegfa 164* is the predominant isoform in all time-points analysed. However, it remains to be determined what is the role of EC-derived *Vegfa* both during development and at the adult stage. Previous results from different authors on the effects in the lung of EC-specific *Vegfa* deletion are conflicting. On one hand, Lee et al (2007) reported that non-inducible *Vegfa* EC-specific knock out resulted in premature death and, in the case of surviving mice, these exhibited lungs with increased chronic inflammation and fibrosis, EC rupture, and collapsed lumen (Lee et al, 2007). On the other hand, Ellis et al (2020) reported that the EC-specific deletion of *Vegfa* at P3 elicits no defects in lung alveolar morphology at P9 (Ellis et al, 2020). The reasons for this discrepancy are unclear. It is possible that the *Vegfa* deletion in ECs is compensated by the endogenous or ectopic expression of *Vegfa* in AT1 cells in the latter mouse model.

Another intriguing observation is that *Vegfa* expression in alveolar AT1 and ECs keeps increasing after birth. In many tissues, the formation of new blood vessels usually occurs in response to hypoxia. Under low oxygen tension, hypoxia-inducible factors are protected from proteolytic degradation, become stable, and activate the expression of a myriad of transcriptional targets, among which is *Vegfa* (Krock et al, 2011). Yet, after birth, lung alveoli become filled with oxygen-rich air. This suggests that *Vegfa* expression in the postnatal lungs must be independent of hypoxia. Understanding the

mechanisms that allow the uncoupling between hypoxia and *Vegfa* levels in postnatal lungs deserves attention in the future.

In addition to the previously described *Vegfa* isoforms, we have identified a novel *Vegfa* isoform that has undergone splicing of all introns except of intron 5, to which we called *Vegfa i5*. This isoform is expressed during lung development both in ECs and AT1 cells. Intron retention is a widespread phenomenon shown to have a functional impact in development, physiology, and disease (Wong et al, 2016; Jacob & Smith, 2017; Yue et al, 2020). According to literature on transcripts with these characteristics, there are three main possibilities: (i) the spliced transcript containing intron 5 is targeted for nonsense-mediated mRNA decay, however, if this mRNA species were rapidly eliminated after its production, the probability of detecting it through RNAseq would be very low, so this hypothesis is unlikely; (ii) it corresponds to a pre-spliced mRNA with a retained intron, whose excision from the pre-spliced mRNAs has been shown to allow a rapid production of the mature mRNA in response to extracellular stimuli (Mauger et al, 2016; Jacob & Smith, 2017); (iii) in rarer cases, transcripts containing premature stop codons may encode truncated proteins (Jacob & Smith, 2017). In this case, the *Vegfa* intron 5 contains multiple premature stop codons, thus, the putative protein codified by *Vegfa i5* should contain exon 1–5 only. According with what has been described (Peach et al, 2018a), it would be able to bind to VEGFRs (contains exons 3 and 4) but not to NRP1 or to the extracellular matrix (does not contain exons 6 and 7). It will be relevant to identify how the generation of this novel *Vegfa* mRNA isoform is controlled, and to address its functional significance during lung development. In addition, it has been described that gene expression changes and RNA polymerase II elongation rate may be coupled with changes in AS (Bentley, 2014). Because we found that both *Vegfa* gene expression and its AS pattern change during lung development, it will be interesting to address if the occurrence of these events on *Vegfa* is interdependent.

In sum, our work contributes for a better understanding of the mechanisms driving alveologenesis and lung adaptation to breathing and sets the ground for the study of the role of AS dynamics during lung development.

# Materials and Methods

### Mice and sample collection

Wild-type C57Bl/6 mice (*Mus musculus Linnaeus*) were used in this study. Both female and male mice were used for sample collection.

**Primers used in standard end-point PCR.**

| Target | Sequence | Reference |
|---|---|---|
| *Vegfa* (E3/E4) | **F1: 5'**-CGACAGAAGGAGAGCAGAAGT-**3'** | This article |
| | **R1: 5'**-ACTCCAGGGCTTCATCGTTA-**3'** | |
| Universal *Vegfa* | **F2: 5'**-CCGAAACCATGAACTTTCTGC-**3'** | This article |
| | **R2: 5'**-GGATTAAGGACTGTTCTGTCAACG-**3'** | |
| *Vegfa* Intron 5 (a) | **F2: 5'**-CCGAAACCATGAACTTTCTGC-**3'** | This article |
| | **R3: 5'**-CCTTCACTGCACGTTTAGACC-**3'** | |
| *Vegfa* Intron 5 (b) | **F3: 5'**-GGAAGGTCAGTTTAGGACGG-**3'** | This article |
| | **R2: 5'**-GGATTAAGGACTGTTCTGTCAACG-**3'** | |
| Gene_AS Event ID | Sequence | Reference |
| *Col13a1*_MmuEX0012150 | **F: 5'**-GTGGAGAGTACCCACACAGG-**3'** | This article |
| | **R: 5'**-TTGGATGCCAGTCTGACTTT-**3'** | |
| *Palm*_MmuEX0033522 | **F: 5'**-GGATCCACAATGATGAAAGC-**3'** | This article |
| | **R: 5'**-GCCTTGTGAATGAGTTCGTC-**3'** | |
| *Arfip1*_MmuEX0005740 | **F: 5'**-TCATGGCTTTGACAGTACCA-**3'** | This article |
| | **R: 5'**-CTGGTCCTCCTTTTGTGTGT-**3'** | |
| *Plod2*_MmuEX0035870 | **F: 5'**-GGGTACTATGCTCGCTCTGA-**3'** | This article |
| | **R: 5'**-TATCAGCCGTCCAAATTCAT-**3'** | |
| *Afdn*_MmuEX0029239 | **F: 5'**-CAACAAGATGGTGAGCATGA-**3'** | This article |
| | **R: 5'**-ATGCCATCTGAACCAAGTGT-**3'** | |
| *Polr2b*_MmuINT0123847 | **F: 5'**-GCAATTGCTAACACGAGGAC-**3'** | This article |
| | **R: 5'**-TGGCAGCTAAACACTCATCA-**3'** | |
| *Hdac7*_MmuEX0022512 | **F: 5'**-CCCAGTAGTAGCAGCACACC-**3'** | This article |
| | **R: 5'**-GAGGGCCTAAAGTTGAATGG-**3'** | |
| *Slain2*_MmuEX0042523 | **F: 5'**-GCCATCCCACAGATTTACAG-**3'** | This article |
| | **R: 5'**-TTCTTGGCTACCAGAACCAG-**3'** | |
| *Rbm26*_MmuEX0039116 | **F: 5'**-GCCCAGAGTGCTACTTCAGA-**3'** | This article |
| | **R: 5'**-TCAACTGCAGGGGTAGAAAC-**3'** | |
| *Mbnl1*_MmuEX0028097 | **F: 5'**-TGGATTACATCAAGGGGAGA-**3'** | This article |
| | **R: 5'**-CATGTTGGCTAGAGCCTGTT-**3'** | |
| *Macf1*_MmuEX0027473 | **F: 5'**-ACATCACAGCTCCTGATTCC-**3'** | This article |
| | **R: 5'**-TCCTCCATTCTGCTGAAGAC-**3'** | |
| *Col6a3*_MmuEX0012337 | **F: 5'**-GACTCCTCCCTGGTCTTCAT-**3'** | This article |
| | **R: 5'**-GGAACTGACCCAAGACATTG-**3'** | |
| *Papola*_MmuEX0033600 | **F: 5'**-TCAGGAAACACAGCAACAAA-**3'** | This article |
| | **R: 5'**-TACCTGAGAGGCCAACAGAG-**3'** | |
| *Myo9b*_MmuEX0030598 | **F: 5'**-AGCTACCAAGAGGAGCCAGT-**3'** | This article |
| | **R: 5'**-CAGACTGAGGGACTGGAGAA-**3'** | |
| *Bnip2*_MmuEX0008099 | **F: 5'**-TGCTAAAGAAACGAGTTGTTGA-**3'** | This article |
| | **R: 5'**-TGGTGGTTCTTGTTTTCCAT-**3'** | |
| *Crlf1*_MmuEX0012750 | **F: 5'**-CCTCTGTTGCTCTGTGTCCT-**3'** | This article |
| | **R: 5'**-CGTGAGATCCTTCATGTTCC-**3'** | |
| *Kif13a*_MmuEX0025416 | **F: 5'**-CATGGTTGAAGCCATCCTAT-**3'** | This article |
| | **R: 5'**-TTCCCGTCGTTTAATGAGTG-**3'** | |

**Continued**

| Target | Sequence | Reference |
|---|---|---|
| *Psd*_MmuINT0127657 | **F: 5'-**AGGCCTACCTGGAGTTTGAG**-3'** | This article |
| | **R: 5'-**TCTATGGTGTCCAGCTCCTC**-3'** | |
| *Nexn*_MmuINT1024226 | **F: 5'-**GTTTGAACAAATGGCAAAGG**-3'** | This article |
| | **R: 5'-**AGTGGAGCCATTAACAATGC**-3'** | |
| *Pecam1*_MmuEX0034408 | **F: 5'-**GCTCCACTTCTGAACTCCAA**-3'** | This article |
| | **R: 5'-**GGGAGGACACTTCCACTTCT**-3'** | |

Timed matings were performed and mouse lungs were collected at different time-points for different procedures. Biological replicates of each time-point were collected from the same litter.

### RNA extraction from bulk lungs

Immediately after isolation, the lungs were transferred to an Eppendorf tube, snap frozen in liquid nitrogen, and stored at –80°C. The lungs were thawed and mechanically disrupted using a pestle. To extract and purify RNA, we used the RNeasy Plus Mini Kit (QIAGEN), following the manufacturers' instructions. For further dissociation of the tissue, after RLT plus lysis buffer addition, we used the pestle and a 20-G needle on a 3-ml syringe (passing the tissue roughly 10 times). RNA concentration was measured using Thermo Fisher Scientific NanoDrop 2000. A fraction of the purified RNA was used to produce cDNA and the remaining volume of RNA was stored at –80°C.

### Lung tissue dissociation and FACS

For cell sorting, the lungs were collected and transferred to a 50-ml falcon with 1 ml of cold (+4°C) DMEM supplemented with 1% BSA. Disruption of the tissue was performed on ice using sterile scissors and a sterile scalpel blade until no clear tissue pieces were visible. The dissociated lung tissue was incubated with 2 ml of Liberase (200 $\mu$g/ml) and DNase I (10 $\mu$g/ml) in DMEM + 1% of BSA in an Eppendorf tube and incubated in a rotator for 40 min at 37°C. During this enzymatic digestion period, the tissue was further disrupted mechanically by passing 4 times through a 20-G needle on a 2-ml syringe. We added 2 ml of cold FACS buffer (EDTA 1 mM, BSA 0.5%, and dPBS 1×) and aspirated the suspension into a 10-ml syringe through a 20-G needle. Cell suspension was forced to pass through a 70-$\mu$m cell strainer. We centrifuged the samples at 200$g$ for 5 min at 4°C. The supernatant was discarded. To lyse RBCs, each pelleted lung was re-suspended in 1 ml of RBC lysis buffer 1× and incubated for 5 min at RT. Cell suspension was forced to pass through a 40 $\mu$m cell strainer. The number of cells was counted and the cell suspension was centrifuged at 200$g$ for 5 min at 4°C. The pelleted cells were re-suspended in FACS buffer to a density of 16 × 10$^6$ cells/ml. Incubation with antibodies and Live/Dead Fixable Viability dye was performed for 20 min in the dark at +4°C in the following dilutions: Live/Dead APC-Cy7 (1:2,000); Rat anti-Mouse APC CD31 (1:100); Mouse

anti-Mouse FITC CD45 (1:200); Rat anti-Mouse PE CD326 (Ep-CAM) (1:100); and Rat anti-Mouse PerCP/Cy5.5 MHC II (1:500). Single color and unstained controls were used to set up the gatings.

After antibody incubation, cells were centrifuged at 300$g$ for 5 min at 4°C and washed by re-suspending in FACS buffer to a concentration of 1 × 10$^6$ cells/200 $\mu$l and centrifugation at 300$g$ for 5 min at 4°C. Finally, pelleted cells were re-suspended in FACS buffer to a final concentration of 1 × 10$^6$ cells/250 $\mu$l for the single colors, and unstained fraction, and 8 × 10$^6$ cells/ml for the fraction to be sorted. FACS was performed in BD FACSAria III cell sorter with a nozzle of 100 $\mu$m and a pressure of 20 PSI. Dead cells were excluded using Live/Dead Fixable Viability dye. We collected different cell populations, in RNase-free Eppendorf tubes filled with 500 $\mu$l of collection buffer (Hepes 25 mM, BSA 2.5% in DMEM) and centrifuged at 2,400$g$ for 5 min at 4°C. The supernatant was discarded and the pellet was re-suspended in 1 ml of Trizol Reagent by pipetting up and down several times and vortexed for 1 min. After homogenization, cells were incubated at RT for 5 min to allow the complete dissociation of nucleoprotein complexes. Cell lysates were stored at –80°C until RNA extraction of sorted cells was performed.

### RNA extraction from sorted cells

The samples in 1 ml of Trizol were thawed on ice and incubated at RT for 5 min. 200 $\mu$l of chloroform was added to each Eppendorf. The tubes were shaken by hand for 30 s and incubated at RT for 5 min. After incubation, the tubes were centrifuged at 12,000$g$ for 15 min at 4°C. The upper aqueous phase was carefully transferred to a new RNase-free Eppendorf tube. To precipitate and wash RNA, 1.5 $\mu$l of glycogen and 50 $\mu$l of 3 M sodium acetate were added to each tube. Tubes were briefly vortexed and 500 $\mu$l of isopropanol was added. The Eppendorf tubes were vortexed and incubated at RT for 15 min and centrifuged at 12,000$g$ for 8 min at 4°C. The supernatants were discarded and the pellets were washed with 1 ml of 75% ethanol. The tubes were centrifuged at 12,000$g$ for 5 min at 4°C and the supernatant was carefully discarded after centrifugation. The pellets were allowed to dry at RT and were resuspended in 20 $\mu$l of RNase-free water. After suspension, all samples were kept on ice and RNA was quantified using Thermo Fisher Scientific NanoDrop 2000. Samples were treated with recombinant DNase I (RNase-free). For each reaction, 5 $\mu$l of DNase I buffer, 1 $\mu$l of DNase I, and 24 $\mu$l of RNase-free water were added to 20 $\mu$l of RNA (up to 10 $\mu$g) from each sample

**Primers used in RT-qPCR.**

| Gene | Sequence | Reference |
|------|----------|-----------|
| Actb | **F: 5'-**CACCCGCGAGCACAGCTTCT-**3'** | This article |
| | **R: 5'-**CGTTGTCGACGACCAGCGCA-**3'** | |
| Gusb | **F: 5'-**AACCTCTGGTGGCCTTACCT-**3'** | This article |
| | **R: 5'-**TCAGTTGTTGTCACCTTCACCT-**3'** | |
| Vegfa 120 | **F4: 5'-**GCCAGCACATAGGAGAGATGAGC-**3'** | This article |
| | **R4: 5'-**GGCTTGTCACATTTTTCTGGC-**3'** | |
| Vegfa 164 | **F4: 5'-**GCCAGCACATAGGAGAGATGAGC-**3'** | This article |
| | **R5: 5'-**CAAGGCTCACAGTGATTTTCTGG-**3'** | |
| Vegfa 188 | **F4: 5'-**GCCAGCACATAGGAGAGATGAGC-**3'** | This article |
| | **R6: 5'-**AACAAGGCTCACAGTGAACGCT-**3'** | |
| Vegfa Intron 5 | **F5: 5'-**CAGATGTGAATGCAGACCAA-**3'** | This article |
| | **R7: 5'-**ACCCAAGAGAGGAAGCAAGA-**3'** | |
| Total Vegfa | **F1: 5'-**CGACAGAAGGAGAGCAGAAGT-**3'** | This article |
| | **R1: 5'-**ACTCCAGGGCTTCATCGTTA-**3'** | |
| CD31 | **F: 5'-**ACACCTGCAAAGTGGAATCA-**3'** | This article |
| | **R: 5'-**CTGGATGGTGAAGTTGGCTA-**3'** | |
| CD45 | **F: 5'-**GGGTTGTTCTGTGCCTTGTT-**3'** | This article |
| | **R: 5'-**CTGGACGGACACAGTTAGCA-**3'** | |
| EpCam | **F: 5'-**TGTCATTTGCTCCAAACTGG-**3'** | This article |
| | **R: 5'-**GTCGTACAGCCCATCGTTGT-**3'** | |
| Aqp5 | **F: 5'-**CCGAGCCATCTTCTACGTG-**3'** | This article |
| | **R: 5'-**TGGTGTTGTGTTGTTGCTGA-**3'** | |
| Sfptc | **F: 5'-**ATGGACATGAGTAGCAAAGAGG-**3'** | This article |
| | **R: 5'-**GATGAGAAGGCGTTTGAGGT-**3'** | |
| Foxj1 | **F: 5'-**GAGCTGGGGACAGAGAACC-**3'** | This article |
| | **R: 5'-**CTCCTCCGAACACGAATGT-**3'** | |
| Prox1 | **F: 5'-**AGAGAGAGAGAAAGAGAGAGAGTGG-**3'** | This article |
| | **R: 5'-**TGGGCACAGCTCAAGAATC-**3'** | |
| Cdh5 | **F: 5'-**GAACGAGGACAGCAACTTCACC-**3'** | This article |
| | **R: 5'-**GTTAGCGTGCTGGTTCCAGTCA-**3'** | |
| Cdh1 | **F: 5'-**TGCCACCAGATGATGATACC-**3'** | This article |
| | **R: 5'-**GCTGGCTCAAATCAAAGTCC-**3'** | |
| Car4 | **F: 5'-**ACAAGGTGAACAAGGGCTTC-**3'** | This article |
| | **R: 5'-**CATGTCCTGCAAACTGCTCT-**3'** | |
| Cpeb4 | **F: 5'-**TCAAGTCCAACCATCAAGGA-**3'** | This article |
| | **R: 5'-**TCGATTCCAGCATAGCAGAC-**3'** | |
| Elavl2 | **F: 5'-**TTGGGTTACGGATTTGTGAA-**3'** | This article |
| | **R: 5'-**CTCCTTCTGGGTCATGGTTT-**3'** | |
| Hnrnpa1 | **F: 5'-**GACAGAGGCAGTGGGAAAA-**3'** | This article |
| | **R: 5'-**AGCCATCTCTTGCTTCGAC-**3'** | |

making a final volume of 50 μl per reaction and incubated for 20 min at 30°C. To inactivate DNase I and purify the RNA, we added one volume of phenol–chloroform–isoamyl alcohol mixture (25:24:1). The tubes were vortexed and centrifuged at 12,000*g* for 10 min at 4°C. The upper aqueous phase was transferred to a new RNase-free Eppendorf tube. 1 volume of chloroform was added to each sample, the samples were then vortexed and centrifuged at 12,000*g* for 10 min at 4°C. The upper aqueous phase was transferred to a new RNase-free Eppendorf. To precipitate RNA, we added 1.5 μl of glycogen and 50 μl of 3 M sodium acetate to each tube and vortexed them. 500 μl of isopropanol was added, the tubes were vortexed and incubated at RT for 15 min. After incubation, the tubes were centrifuged at 12,000*g* for 20 min at 4°C to precipitate RNA and the supernatant was discarded. To wash the pellets, we added 1 ml of 75% ethanol and centrifuged the tubes at 12,000*g* for 5 min at 4°C. We discarded the supernatant carefully and repeated this step once to further wash the RNA. Pellets were dried at RT and re-suspended it in 15 μl of RNase-free water. Each sample was kept on ice and quantified using Thermo Fisher Scientific NanoDrop 2000. A fraction of the purified RNA was used to produce cDNA and the remaining volume of RNA was stored at −80°C.

### Production of cDNA

Production of cDNA from RNA extracted from both bulk lungs and sorted cells was performed using the High-Capacity RNA-to-cDNA Kit (Applied Biosystems), following the manufacturers' protocol. The cDNA was stored at −20°C and used for end-point PCR and RT-qPCR reactions.

### End-point PCR

Specific target regions were amplified through standard end-point PCR. Primers used to amplify the distinct *Vegfa* isoforms through end-point PCR are presented in the table below. PCR was performed using Q5 Hot Start High-Fidelity DNA Polymerase. To perform each reaction we added 10 μl of 5X Q5 reaction buffer, 1 μl of dNTPs (10 mM), 2.5 μl of forward primer (10 μM), 2.5 μl of reverse primer (10 μM), 0.5 μl of Q5 Hot Start High-Fidelity DNA Polymerase, cDNA as template and nuclease-free water up to 50 μl. PCR was run on T100 Thermal Cycler (Bio-Rad) using the following PCR program: an initial denaturation step of 98°C for 30 s, 26 cycles of 10 s at 98°C, 30 s at 65°C, and 2 min at 72°C, followed by a final extension of 2 min at 72°C. PCR products were analysed by performing electrophoresis in a 2% agarose gel or in a TBE-UREA-polyacrylamide gel, as described below.

To validate a fraction of AS events identified on RNAseq, 20 specific target regions were amplified through end-point RT-PCR on cDNA from triplicate samples of bulk mouse lungs at E18.5 and P5 (AS events ID and primers used are presented on the table below). PCR was performed using GoTaq G2 Flexi DNA Polymerase, according to the manufacturers' instructions.

PCR was run on T100 Thermal Cycler (Bio-Rad). The PCR program was adapted to the characteristics of each primer pair and PCR product, as follows: an initial denaturation step of 95°C for 2 min, 32–40 cycles of 40 s at 95°C, 34 s at 59–61°C and 15–55 s at 72°C, followed by a final extension of 5 min at 72°C. PCR products were analysed by performing electrophoresis in a 2% agarose gel or in a

TBE-UREA-Polyacrylamide gel as described below. The relative intensity of the bands obtained was quantified using Fiji/ImageJ software and used to calculate the ΔPSI for each gene.

### PCR product purification and Sanger sequencing

To sequence the *Vegfa* isoform comprising intron 5, *Vegfa i5*, several end-point PCR reactions using different primer pairs were performed to obtain partially overlapping DNA fragments covering all the *Vegfa i5* sequence. From the total volume of each PCR product, we saved a fraction (10 μl) to run on a 2% agarose gel to confirm if the amplification of each fragment was successful. The remaining volume (40 μl) was purified using the Wizard SV Gel and PCR Clean-Up System (Promega) following the manufacturers' protocol. DNA Sanger sequencing was performed by GATC services (Eurofins). DNA sequences obtained from the sequencing results of fragments amplified from *Vegfa i5* (FASTA file) were imported into the SnapGene software (GSL Biotech).

### Gene expression analysis using real-time quantitative PCR (RT-qPCR)

Quantitative gene expression analysis was performed by RT-qPCR using intron-spanning primer pairs. Two housekeeping genes, *Actb* and *Gusb*, were analysed as housekeeping genes. All primers used for RT-qPCR are indicated in the table below. A standard curve for each primer pair was obtained in every RT-qPCR run alongside with the samples to be analysed. To obtain the standard curve, we mixed cDNA from all time-points collected and prepared different dilutions (1:10, 1:100, 1:500, 1:1,000).

In addition, in each plate, we used a calibrator sample. The calibrator sample, which was prepared once and used in every plate, was generated by mixing equal quantities of each cDNA sample (all time-points) from bulk lungs (1:200). For each individual reaction, we added 7 μl of Power SYBR Green PCR Master Mix, 0.3 μl of previously diluted primer pairs (final concentration of 100 nM), 2 μl of diluted cDNA, and 4.85 μl of RNase-free water resulting in a final volume of 14 μl per well. We used Applied Biosystems VIIA 7 Real-Time PCR system, the conditions for the reaction were 1 × 50°C for 2 min, 95°C for 10 min; 45 × 95°C for 15 s, 60°C for 1 min, 1 × 95°C for 15 s, 60°C for 1 min; and 1 × 95°C for 15 s. The software used to analyze each RT-qPCR experiment was QuantStudio Real-Time PCR Software (Applied Biosystems). Standard curves, Melting curve and amplification plots were all generated in this software. Pfaffl method (Pfaffl, 2001) was used to quantify gene expression. To avoid inter-plate variations, when we needed to compare across different plates, we used an adaptation of the method used in qbase+ software (Biogazelle) that allows relative quantification of gene expression through a modified method based on ΔΔCt and through the normalization to a calibrator sample (Calibrated Normalized Relative Quantity, CNRQ). All graphics presented were elaborated with Graph Pad Prism 8 software.

### TBE-urea-PAGE

6% of TBE-urea-polyacrylamide gels were prepared by using: 1.5 ml of 10× TBE buffer, 2.25 ml of 40% polyacrylamide/bisacrylamide

reagent, 7.2 g of urea, and ddH$_2$O up to 15 ml. After completely dissolving the urea, we added 15 $\mu$l of TEMED and 150 $\mu$l of fresh 10% (wt/vol) APS. The mixture was poured into a 15-mm-thick gel support and a 15-well comb was inserted. After polymerization, we mounted the gels in an electrophoresis apparatus filled with 1× TBE buffer. Before loading the samples, urea traces and gel pieces were washed from the wells with 1× TBE Buffer and the gel was pre-ran for 30 min at 25 V. We loaded 30 $\mu$l of sample in each well: 15 $\mu$l of PCR product diluted in 15 $\mu$l of homemade 2× TBE-urea sample buffer (10% 10× TBE running buffer; 6% Ficoll Type 400; 1% bromophenol blue; 7 M urea). The gel was run at 25 V until the samples moved past the wells into the gel, after which the voltage was increased to 100 V. After running, the gel was incubated in 1× TBE for 10 min and after stained with Green Safe reagent dye (5 $\mu$l/100 ml of TBE 1×) for 30 min and visualized on ChemiDoc XRS+ system (Bio-Rad).

### Cytospin

Single-cell suspensions from lungs at P5 before sorting (pre-sort sample) and after sorting were used in cytospin (Koh, 2013). 80,000–100,000 cells/200 $\mu$L FACS buffer were used per slide. Samples were centrifuged in Shandon Cytospin 2 for 5 min at 500$g$. The resultant slide was fixed with 4% PFA for 10 min at RT, washed twice in PBS, and stored in PBS 0.01% Azide at +4°C until further use.

### Lung explants culture

E17.5 mice were euthanized, the fetal mouse lungs dissected free of surrounding structures, and washed in ice-cold HBSS supplemented with 50 units/ml of penicillin–streptomycin. The lung tissue was minced into 1–3 mm$^3$ cubes using a sterile blade, and mouse lung explants were cultured on an air–liquid interface using Transwell cell culture inserts of 12 mm diameter, PES membrane, 0.4 $\mu$m pore size (Corning, 3460). Explants were cultured for 24 h in DMEM/F-12, 50 units/ml of penicillin–streptomycin, and 0.25 mg/ml ascorbic acid at 37°C in 95% air/5% CO2, supplemented with control, recombinant mouse VEGFA 164 or VEGFA 188 at a final concentration of 5.26 nM.

### Immunofluorescence

For immunostaining in cytospin slides (all steps at RT), slides were blocked in 3% BSA 0.1% triton X-100 in PBS (PBST-BSA) for 30 min. Primary antibody incubation was performed in PBST-BSA for 2 h. Slides were washed three times in PBS 0.1% Triton X-100 (PBST). Secondary antibody incubation was performed in PBST-BSA for 1 h. Slides were washed three times in PBST. Nuclei were stained with DAPI (1:10,000) for 5 min. Slides were washed once with PBS and mounted in Mowiol/Dabco mixture.

For immunofluorescence of mouse lung explant slices, cultured mouse lung explants were collected, washed in PBS and fixed in 2% PFA in PBS for 30 min at +4°C on a swinging platform. After fixation, explants were washed in PBS overnight at +4°C on a swinging platform. For cryopreservation, the explants were incubated PBS with 20% (wt/vol) sucrose and 33% (vol/vol) optimal cutting temperature compound (OCT) in PBS for 48 h at +4°C on a rotator platform, after which were embedded in OCT and frozen. OCT explant blocks were cut into 10 $\mu$m slices using a Cryostat Leica CM3050S, mounted on SuperFrost Plus slides (Thermo Fisher Scientific), and stored at –20°C until further use.

For immunofluorescence, lung slices were rehydrated in PBS for 10 min. Lung slices were blocked for 30 min at RT in blocking buffer (5% FBS, 0.5% Triton X-100 in PBS). Primary antibodies incubation was performed in blocking buffer on a humidified chamber overnight at +4°C. After three washes with PBS for 15 min each, sections were incubated with secondary antibodies in blocking buffer for 60–90 min at RT. After three washes with PBS for 15 min each, slices were counterstained with DAPI (1:10,000) for 5 min, washed once in PBS, and mounted using a Mowiol/Dabco mixture.

The antibodies used in immunofluorescence were: Rabbit Pro-SFPTC (1:1,000; SevenHills BioReagents), Goat CD31 (PECAM1) (1:400; R&D), Rabbit Aquaporin 5 (1:200; Merck), Rabbit ERG (1:200; Abcam), Goat Car4 (1:150; R&D), Rat E-cadherin (ECCD-2) (1:1,000; Invitrogen), Donkey anti-Rabbit Alexa 568 (1:500; Thermo Fisher Scientific), Donkey anti-Goat Alexa 647 (1:500; Thermo Fisher Scientific), Donkey anti-Goat Alexa 555 (1:400; Thermo Fisher Scientific), and Donkey anti-Rabbit Alexa 488 (1:500; Thermo Fisher Scientific).

Images were acquired on a Zeiss LSM 710 inverted laser scanning confocal microscope (Carl Zeiss) equipped with a motorized stage, using an EC Plan-Neofluar DIC 40×/1.3 oil objective or on a Zeiss LSM 980 inverted laser scanning confocal microscope (Carl Zeiss) equipped with a motorized stage, using a Plan-Apochromat 20×/0.8 dry objective. Image analysis was performed with Fiji/ImageJ software.

### ELISA

After collection, lungs were snap-frozen in liquid nitrogen. Ice-cold lysis buffer (150 mM NaCl; 1 mM EDTA; 50 mM Tris–HCl pH = 7.4; 1% Triton X-100 diluted in ddH$_2$O and supplemented with 1 mM DTT and proteinase and phosphatase inhibitors [1861282; Thermo Fisher Scientific]) was added to the Eppendorf tube with the frozen lungs. Volume of lysis buffer was adjusted according to lung size (500 $\mu$l/150 mg). To mechanically disrupt the tissue, we used a pestle and pipetted up and down. We incubated the mixture on ice for 15 min and centrifuged at maximum speed for 15 min at +4°C. We transferred the supernatant to a new ice-cold Eppendorf tube. To quantify protein concentration, we used the Pierce BCA Protein Assay Kit following the manufacturers' protocol. For analysis of VEGFA protein levels, we used the Mouse VEGFA Quantikine ELISA Kit (MMV00; R&D), following the manufacturers' protocol.

### RNAseq and bioinformatics analysis

#### RNA isolation and sequencing library preparation
We performed RNAseq using mouse lungs at four developmental stages (E15.5, E18.5, P5, and P8) in triplicates. After RNA extraction, RNA integrity was evaluated in Fragment Analyzer and all RNA samples revealed to have an RNA quality number >9.8. Library preparation was performed by using Truseq RNA Library protocols. Samples were barcoded, pooled and redistributed into three lanes. RNA sequencing was performed using HiSeq 4000 sequencing

platform at BGI Genomics. In total, we obtained 897 million paired-end (PE) 101-nt reads (59.8 million per sample on average) (Fig S1A).

### RNAseq data alignment and differential gene expression

The ~56.8 M raw reads generated per sample were uniquely mapped to the mm10 assembly and annotated to the Gencode vM14 transcriptome using TopHat2 (version 2.1.1) (Kim et al, 2013), with Bowtie2 (version 2.3.4) (Langmead & Salzberg, 2012).

For gene expression analysis, gene expression levels were quantified with HTSeq-count (version 0.10.0) (Anders et al, 2015). Data preprocessing was done in R, using *limma* and *edgeR* packages, as follows: genes were removed when weakly expressed or associated to noninformative features ("no_feature," "ambiguous," "too_low_aQual," "not_aligned," "alignment_not_unique"); and for features without at least one read per million in three of the samples (minimum number of replicates). After applying these quality criteria, we obtained 16,152 expressed genes for downstream analysis. The counts per gene were normalized to counts per million (CPMs) by dividing it by the total number of mapped reads per sample and multiplying by 10^6. The CPM normalized data were then transformed with $\log_2$ using an offset of one. Pairwise analysis of differential gene expression was performed using the generalized linear model workflow (Robinson et al, 2010) and the cutoffs $|\log_2(FC)| > 1$ and FDR < 0.05. Row z-scores for each gene represented in the heat maps were calculated using $\log_2$ (CPMs + 1) values and using *heatmap.2* function from gplots package in R.

### AS analysis

For AS event-level analysis, raw reads were mapped to previously annotated AS events for the mouse reference assembly (mm10 vastdb.mm2.23.06.20) from VastDB (VastDB v2 released). Abundance of AS events in PSI values was estimated using VAST-TOOLS (version 2.5.1) *align* and *combine* commands (Irimia et al, 2014; Tapial et al, 2017).

AS events in which the read coverage based on corrected reads (quality score 2) does not meet the minimum threshold (N) and those in which PSI could not be determined (NA) in at least one sample, and those in which minimum number of reads < 10 in at least three samples were excluded from subsequent analysis. To calculate differential AS between each pair of time-points for the AS events, we used VAST-TOOLS *diff* command. Vast *diff* uses Bayesian inference followed by differential analysis to calculate ΔPSI and minimum value difference (MV) between two samples for each AS event. We used a confidence interval of 95%, MV ≥ 10%. After these criteria, we obtained 460 events identified as being differentially AS in at least one pairwise comparison between time-points, associated with 355 genes (Table S1).

To analyze the expression levels of genes undergoing differential AS in at least one pairwise comparison between time-points, we divided the 16,152 expressed genes into four bins of equal size according to their absolute level of expression (CPMs) (low expression, medium-low expression, medium–high expression, and high expression). Then, we distributed the genes undergoing differential AS in at least one pairwise comparison between time-points (355 genes) within these bins.

To select genes that undergo differential AS in at least one pairwise comparison between time-points but do not undergo

changes in gene expression in the same time interval, we used a confidence interval of 95% with MV ≥ 10% (differential AS) and $|\log_2(FC)| > 1$ with FDR < 0.05 (differential gene expression). After this cutoff, we obtained 371 AS events associated with 295 genes.

Clustering of AS events into distinct sets based on their kinetics along lung development was performed by two methods: K-means and hierarchical clustering. K-means clustering was performed using the *kmeans* function (amap package) in R using Spearman correlation as a correlation method for each pair of AS events based on their PSI values at the different time-points. The average silhouette width method was used to find the best number of clusters, using the *fviz_nbclust* function (factoextra package) in R. The AS events included in each cluster are listed in Table S5. Hierarchical clustering was performed using *hclust* function (stats package) in R using Spearman correlation as a correlation method for each pair of AS events based on their PSI values at the different time-points and the complete-linkage method as clustering method.

Heat maps of clustered AS events were represented in heat maps using the *heatmap.2* function in R. For the heat map representation, logit (PSI) values were used and were scaled by row by calculating row z-scores.

For transcript-level analysis, transcripts isoform abundance in RNAseq datasets in transcripts per million (TPM) was estimated using Kallisto (version 0.44.0) (Bray et al, 2016). *Vegfa i5* isoform was manually annotated in Kallisto index built with reference transcriptome before performing the pseudo-alignment.

### Pathway analysis

KEGG pathway analysis was carried out using DAVID v6.8 (Huang et al, 2009a, 2009b). KEGG terms with a modified Fisher exact *P*-value (EASE score) < 0.05 and FDR < 0.1 were considered. The set of 16,152 genes we identified to be expressed in the lungs during the time-points analysed was used as control background list.

### RBP motif enrichment analysis

To identify potential RPBs regulators of the AS events detected during lung development, we used Matt, a unix toolkit that searches for RBPs motifs in genomic sequences (Ray et al, 2013). Briefly, we used the *get_vast* command to extract subsets of reported AS events generated by VAST-TOOLS. We considered only AS events in non-differentially expressed genes. Then, we filtered the table for intron retention events (IR) and exon skipping events of type S, C1 and C2, with PSI values with a minimum quality flag of LOW. We defined categories for each event (enhanced or silenced) based on the ΔPSI values obtained from VAST-TOOLS (confidence interval of 95% with MV ≥ 10%). A set of unregulated events was randomly selected from the same genes with significant AS events to be used as control in the enrichment analysis. Upstream and downstream sequences of each event were obtained with Matt's *get_seqs* command, comprising the 250 nt flanking each splice site (150 nt towards the intron and 100 nt towards the exon). Then, we performed an enrichment test using Matt's *test_cisbp_enrich* function, which compares the positional density of motifs from the database (Ray et al, 2013) in enhanced or silenced AS events against the background set of unregulated exons or introns, predicting the number of hits between the two groups of sequences associated with each RBP motif (Gohr & Irimia, 2019). The Matt function performs a

permutation test to determine significant enrichment/depletion of RBP motifs ($P$-value < 0.005). Finally, we used a custom python script based on the *regex* package to find the number of events containing hits for each RBP motif, and plotted these values normalized by the total number of events of each type. In addition, we used Matt's function *get_regexp_prof* to plot the positional distributions of motif hits across sequences. We selected RBPs that undergo changes in gene expression between E18.5 and P5 ($|\log_2FC| > 1$ and FDR < 0.05).

To reinforce the results of the putative regulators, we explored the CLIP-seq profiles of 356 RBPs in human cancer cells (HepG2 and K562), previously published and available in ENCODE (Van Nostrand et al, 2020). First, mouse AS events were converted to the respective human homologous exons using VastDB (Tapial et al, 2017), and the equivalent genomic intervals (250 nt flanking each splice site) were crossed with the different CLIP-seq profiles for the enriched RBPs. Individual CLIP-seq profiles for each locus were produced using Bedtools (Quinlan & Hall, 2010) and R (https://www.r-project.org/) (R Core Team, 2022).

### Statistical analysis

Statistical analysis was performed using GraphPad Prism 8. Measurements were taken from distinct samples, and statistical details of experiments are reported in the figures and figure legends. Sample size is reported in the figure legends. The biological replicate is defined as the number of cells, images, explants or animals, as stated in the figure legends. Comparisons between two experimental groups were analysed with two-tailed unpaired $t$ test, whereas multiple comparisons between more than two experimental groups were assessed with one-way ANOVA with Tukey correction for multiple comparisons. Unless otherwise specified, data are represented as mean ± SD; n.s.: $P > 0.05$; *$P < 0.05$, **$P < 0.01$, ***$P < 0.001$, and ****$P < 0.0001$. We considered a result significant when $P < 0.05$.

### Ethics

Animal experimentation: Mice were maintained at the Instituto de Medicina Molecular under standard husbandry conditions and under national regulations, under the license AWB_2015_11_CAF_Polaridade/ex vivo_surplus_not of use. Animal procedures were performed under the DGAV project license 0421/000/000/2016.

# Data Availability

The RNAseq data have been deposited into the NCBI Gene Expression Omnibus (https://www.ncbi.nlm.nih.gov/geo/) under the accession number GSE175403.

# Supplementary Information

# Acknowledgements

ariana Ferreira, Marie Bordone, Nuno Agostinho and Nuno Barbosa Morais (Disease Transcriptomics Lab, iMM) for input on bioinformatics analysis. Pedro Papotto, Karine Serre (Immuno-Biology & Immuno-Oncology Lab, iMM), Idálio Viegas (Biology of Parasitism Lab, iMM), Isabel Alcobia (Institute of Histology and Developmental Biology, FMUL), Debanjan Mukherjee, Vanessa Zuzarte-Luís (Biology & Physiology of Malaria, iMM), and Mahak Singhal (Vascular Oncology and Metastasis Division, DKFZ), for input on experimental protocols. Luís Oliveira (Cell Architecture Lab, iMM) for input on microscopy image analysis. Flow Cytometry, Bioimaging, Rodent and Comparative Pathology Units at iMM for technical support. All members of the Vascular Morphogenesis lab at iMM for discussions, helpful input and for carefully reviewing the manuscript. Nuno Barbosa Morais for carefully reading this manuscript and providing helpful and critical feedback. This work was supported by European Research Council (ERC starting grant [679368]), the European Union (H2020-TWINN-2015 – Twinning [692322]), Fundação para a Ciência e Tecnologia (FCT) (PTDC/MED-PAT/31639/2017, and UIDP/04378/2020 of the Research Unit on Applied Molecular Biosciences - UCIBIO), and Fondation Leducq (17CVD03). CG Fonseca was supported by a PhD fellowship from the doctoral program Bioengineering: Cellular Therapies and Regenerative Medicine funded by Fundação para a Ciência e Tecnologia (FCT) (PD/BD/128375/2017). T Balboni was supported by a PhD fellowship from the doctoral program "Oncology, Hematology and Pathology - 30th Cycle" funded by University of Bologna, Italy. P Caldas was supported by a postdoctoral researcher fellowship from FCT (PTDC/MED-ONC/28660/2017). AASF Raposo was supported by FCT and Fundo Europeu de Desenvolvimento Regional (FEDER) PAC-PRECISE-LISBOA-01-0145-FEDER-016394 and by an assistant researcher contract from FCT (CEECIND/01474/2017). AR Grosso was supported by a principal investigator contract from FCT (CEECIND/02699/2017). FF Vasconcelos was supported by a postdoctoral researcher contract from FCT (CEECIND/04251/2017). CA Franco was supported by a principal investigator contract from FCT (CEECIND/02589/2018).

## Author Contributions

MF Fidalgo: conceptualization, resources, data curation, formal analysis, validation, investigation, visualization, methodology, and writing—original draft, review, and editing.
CG Fonseca: conceptualization, resources, data curation, software, formal analysis, validation, investigation, visualization, methodology, and writing—review and editing.
P Caldas: data curation, software, formal analysis, validation, investigation, visualization, methodology, and writing—review and editing.
AASF Raposo: resources, data curation, software, formal analysis, investigation, and writing—review and editing.
T Balboni: resources, data curation, formal analysis, investigation, and writing—review and editing.
L Henao Mišíková: data curation, formal analysis, investigation, and writing—review and editing.
AR Grosso: data curation, software, formal analysis, validation, investigation, visualization, methodology, and writing—review and editing.
FF Vasconcelos: conceptualization, resources, data curation, software, formal analysis, supervision, funding acquisition, validation, investigation, visualization, methodology, project administration, and writing—original draft, review, and editing.
CA Franco: conceptualization, supervision, funding acquisition, methodology, project administration, and writing—review and editing.

## Conflict of Interest Statement

The authors declare that they have no conflict of interest.

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
