## [Reviewer comments · Life Science Alliance]

Life Science Alliance

Aerocyte specification and lung adaptation to breathing is dependent on alternative splicing changes

Marta Fidalgo, Catarina Fonseca, Paulo Caldas, Alexandre Raposo, Tania Balboni, Lenka Henao Mišíková, Ana Grosso, Francisca Vasconcelos, and Claudio Franco

DOI: <https://doi.org/10.26508/lsa.202201554>

Corresponding author(s): Francisca Vasconcelos, Instituto de Medicina Molecular João Lobo Antunes and Claudio Franco, University Lisbon

Review Timeline:

Submission Date:	2022-06-08
Editorial Decision:	2022-06-13
Revision Received:	2022-06-16
Editorial Decision:	2022-06-27
Revision Received:	2022-09-22
Accepted:	2022-09-22

Transaction Report:

Please note that the manuscript was reviewed at Review Commons and these reports were taken into account in the decision-making process at Life Science Alliance.

June 13, 2022

Re: Life Science Alliance manuscript #LSA-2022-01554

Dr. Francisca F Vasconcelos
Instituto de Medicina Molecular João Lobo Antunes
Vascular Morphogenesis Lab
Avenida Professor Egas Moniz
Lisboa 1649-028
Portugal

Dear Dr. Vasconcelos,

Thank you for submitting your manuscript entitled "Aerocyte specification and lung adaptation to breathing is dependent on alternative splicing changes" to Life Science Alliance. We invite you to re-submit the manuscript, revised according to your Revision Plan.

Thank you for this interesting contribution to Life Science Alliance. We are looking forward to receiving your revised manuscript.

Sincerely,

B. MANUSCRIPT ORGANIZATION AND FORMATTING:

1. General Statement

We would like to thank the reviewers for their careful evaluation of our study and their constructive comments. We have addressed the comments and concerns raised as described below in the revision plan. We believe that the revisions will significantly improve the quality and impact of the manuscript.

To accommodate the new results and display the results in a more harmonious manner, we have redistributed the results displayed in Figures 1, 2, 5, 6, S1, S2, S4 and S6.

All the changes incorporated in the manuscript file are highlighted in blue.

2. Description of the revisions that have already been incorporated in the transferred manuscript

Reviewer #1 (Evidence, reproducibility and clarity (Required)):

This is an interesting manuscript exploring alternative splicing (AS) changes during lung development. AS has been recognized as one of the most important drivers of proteome diversity, with over 94% of genes being alternatively spliced in humans. Splice isoforms may have functions as different as distinct genes; there are many examples of splice isoforms encoding even opposing functions, e.g. pro- and anti-apoptotic. Despite this, the importance of AS in regulating many biological processes, including organ development, has been much less studied than transcriptional regulation for instance. Therefore, I believe this manuscript is timely and important.

In general, the manuscript is well-written, experiments are well designed and executed and proper conclusions drawn based on the data presented. However, some parts are of lower quality, and I think more work is needed.

Some comments/suggestions:

- I am a bit surprised that the authors are describing in quite some detail VEGF isoforms in the Introduction but do not talk about the VEGF "b" antiangiogenic isoforms (see for example PMID 18923433)

Point 1.1 – VEGFA_{xxb} isoforms differ from classical VEGFA_{xxx} isoforms by a different alternative splicing of exon 8 (inclusion of exon 8a or 8b, respectively). VEGFA_{xxb} isoforms have been originally identified in human renal cell carcinoma (Bates et al., 2002). However, the existence of these isoforms remains debatable (Dardente et al., 2020; Harris et al., 2012).

Our analysis presented in Figure 4E has been performed using primers that should also recognise VEGFA_{xxx}b isoforms. However, we only identified 3 bands corresponding to the VEGFA120, 164 and 188 isoforms. Thus, we conclude the VEGFA_{xxx}b isoforms are either not expressed during mouse lung development or are expressed at very low levels.

This is in accordance with the analysis of publically available expressed sequencing tags (ESTs) and RNAseq datasets, which failed to identify expression of VEGFA_{xxx}b isoforms (Bridgett et al., 2017). In addition, in the adult mouse lungs, VEGFA_{xxx}b isoforms have not been detected by RT-PCR (Harris et al., 2012). Therefore, in accordance with our results, there is cumulative data suggesting that VEGFA_{xxx}b isoforms are either not expressed or expressed at very lowly levels in vivo.

However, we understand and agree with the reviewer's concern, and we have now explicitly mentioned these isoforms in the introduction of our manuscript. Also, for the sake of clarity, we have included a sentence in the results section commenting on the expression levels (very low or inexistent) of VEGFA_{xxx}b isoforms during mouse lung development.

- Figure 1 - the depth of sequencing (~60 million per sample) is quite low for splice isoforms analysis, which is usually done closer to 100 million;

- to evaluate how good is the sequencing data, the authors compare with several gene expression reports in the literature - however, they use these data mainly to obtain splicing analysis not gene expression - would be more convincing if their sequencing data is correlated with splice isoforms reported in the literature

Point 1.2 - We agree with the reviewer's comments, by using more than 60 M reads per sample, it would likely reveal the existence of more alternative splicing events, particularly those that are less abundant than the ones we have identified. However, a coverage 60 M reads per sample was sufficient to recover a large number of alternative splicing events. Moreover, in our analysis using Vast-Tools we have only considered those alternative splicing events that have enough coverage to be detected accurately, as stated by the software original publication (<https://github.com/vastgroup/vasttools>). In order to strengthen this point in our MS, we have added the coverage supporting each splicing event in the Supplementary Table S1.

We also agree with the reviewer that a comparison between our and previously published datasets using alternative splicing rather than with gene expression changes would be a more elegant way of validating our dataset. However, to our knowledge, there is no previously published dataset describing alternative splicing changes during mouse lung development. In our literature review, we could only find studies related to lung cancer. Regarding lung development, we could only identify two genes previously described to undergo alternative splicing during lung development: Vegfa, which we carefully examine in this manuscript; and Cav1 (Ramirez et al., 2002). However, the major alternative splicing change of Cav1 was reported to occur between E12.5 and E15.5, a time interval that has not been included in our analysis. Therefore, we are afraid are unable to satisfy the reviewer's request.

- analysis of splice variants by RNAseq may be quite variable depending on the software used (especially if using short-sequencing methods; therefore it is essential to validate a subset of the events detected (ideally

at least 10%) by RT-PCR; additional software may be used as well; depending on available funds and expertise the short sequencing could be backed up by long-sequencing, to be sure important splice events are not missed.

Point 1.3 – We thank the reviewers for the suggestion of further validation. We have proceeded as requested. We have already designed primers for a subset of identified alternative splicing events (N=20) and analysed them by RT-PCR using independent RNA triplicate samples for E18.5 and P5. were able to validate 18/20 of the AS changes analysed and found a strong correlation between the Δ PSI values obtained by the two methods (**new Figure S1C-D**).

- *Figure 2 - the strategy on how the authors found the involvement of RBPs Cpeb4, Elavl2/HuB and hnRNP A1 is great; however, this needs validation by wet lab approaches - e.g. western blot or RT-PCR; also, would be interesting to question their functional importance by manipulating their expression in (at least) some in vitro assays.*

Point 1.4 – We thank the reviewer's suggestions.

We have now examined the gene expression changes of the identified RBPs by RT-qPCR in bulk lungs and in sorted lung cell populations. The analysis of bulk lungs was concordant with what we have described on the RNAseq analysis: Cpeb4 expression increases, while Elavl2 and Hnrnpa1 expression decrease from E15.5 to P8 (**new Figure 2D**).

The expression analysis on sorted cell populations revealed that Cpeb4 increases in expression during lung development on CD31 SP, EpCAM SP and TN cell populations, while Elavl2 and Hnrnpa1 decrease in expression during lung development in EpCAM SP and TN cell populations (**new Figure S5D**).

Remarkably, their highest expression fold changes within the developmental time interval analysed occur in EpCAM SP population (**new Figure S5D**), supporting the hypothesis that the expression changes of these RBPs in EpCAM SP cell population may drive AS changes, such as those observed for Vegfa in this same cell type.

Due to the lack of time, staff and funds, we have not performed the experiments testing the functional impact of the splicing regulators.

- *Figure 3 - the discovery of the VEGFi5 isoform is very interesting - could the authors model/predict how this extra coding region affects the protein structure/function*

Point 1.5 – We thank the reviewer for this comment. In the Discussion section (lines 478-494 of the original submitted version of the manuscript), we had speculated about the possible fates of the Vegfa i5 isoform, but perhaps we were not as clear as we intended. Briefly, the Vegfa intron 5 contains multiple premature stop codons. According to literature, there are 3 main possibilities: *i*) the spliced transcript containing intron 5 is targeted for nonsense-mediated mRNA decay. However, if this mRNA species were rapidly eliminated after its production, the probability of detecting it through RNAseq would be very low; *ii*) it corresponds to a pre-spliced mRNA with a retained intron. The excision of a retained intron from pre-

spliced mRNAs has been shown to allow a rapid production of the mature mRNA in response to extracellular stimuli (Jacob and Smith, 2017; Mauger et al., 2016); *iii*) in rarer cases, transcripts containing premature stop codons may encode truncated proteins (Jacob and Smith, 2017). In this case, the protein codified by VEGFAi5 should contain exon 1-5 only. According with what it is described in the literature (Peach et al., 2018), it would be able to bind to VEGFRs (contains exons 3 and 4) but not to NRP1 or to the extracellular matrix (does not contain exons 6 and 7).

Thus, to accommodate the reviewer's suggestion, we have now further elaborated on these observations to make our ideas clearer regarding the consequences of the possible expression of a truncated VEGFA protein (please see **lines 529-539**).

- Figure 4B - this deals with "all VEGF isoforms" - again, I am surprised that the authors do not show/mention the "b" anti-angiogenic isoforms; it does not matter if they cannot detect them, they might be at low level of expression; they should still be mentioned

Please refer to response provided in **Point 1.1** above.

Overall, I would like to see some more functional implications - splicing changes may be just associated with a process (e.g. lung development) and not causal; just describing the proportions of various VEGF isoforms during lung development does not uncover how they modulate this process.

Reviewer #1 (Significance (Required)):

Describe the nature and significance of the advance (e.g. conceptual, technical, clinical) for the field. and Place the work in the context of the existing literature.

Description of alternative splicing (AS) changes during lung development; how AS contributes to regulation of many biological processes is not very well studied; focus on VEGF isoforms is also important, as many papers describe only how levels of expression of VEGF regulate various processes

- State what audience might be interested in and influenced by the reported findings. splicing biologists, developmental biologists

- Define your field of expertise with a few keywords to help the authors contextualize your point of view. alternative splicing, VEGF biology/ angiogenesis

Reviewer #3 (Evidence, reproducibility and clarity (Required)):

Major findings of the data

Developmental time points of mouse lung bulk RNA were used to identify changes in RNA splicing and to predict potential mediators e.g., *hnRNPA1*, *Cpeb4*, *Elavl2*. Developmental changes in *Vegfa* isoforms were noted and linked to high levels of expression in AT1 cells as the primary source. DAS events were generally independent of the level of transcript; variants increasing and decreasing in the perinatal period were enriched in genes associated with cell adhesion and HIPPO pathways. Findings are consistent with changes in gene expression that occur in the perinatal period of mouse lung development identified previously. RBPs candidates related to splicing or exon skipping were predicted from the analysis. The authors identified transcripts that include exon5 of the *VEGFa*-RNA; termed *Vegfa-i5*. A careful time course of *Vegfa* isoforms was provided. Changes in *Vegfa* occurred during the perinatal period. Major cell types were isolated, including epithelial, endothelial, mesenchymal, and immune cells by standard methodologies, and bulk RNAs profiled focusing on *Vegfa* splicing in epithelial enriched cell fractions using FACs, implicating Type 1 cells as the primary source.

Major comments:

This is a carefully performed, well-written, and well-documented analysis of changes in RNA and RNA splicing during a critical window of perinatal lung development. It is highly descriptive but provides a careful analysis of changes in splicing at a genome-wide level, focusing on changes in *Vegfa* splicing. The authors carefully document developmental changes in *Vegfa* splicing as well as a *VEGFai5* isoform. Standard FACs were used to enrich AT1 and AT2 cells identifying AT1 cells as relatively enriched for *Vegfa* and its isoforms consistent with single-cell data now widely available to investigators. These findings are consistent with a number of published RNA-Seq studies, both bulk and single cell in the mouse where *Vegfa* is developmentally expressed and enriched in alveolar cells including AT1 cells. As such, similar data are available on various websites, for example, HCA, LGEA, JAX Lab that includes both bulk, sorted, and single cell, confirming present studies. These previous studies might be interrogated, compared with present data and carefully referenced.

Point 3.1 – We thank the reviewer for this suggestion. We agree that it is important to mention that it has been previously demonstrated that *Vegfa* is expressed from AT1 cells during lung development. In fact, we have acknowledged that AT1 cells express *Vegfa* (lines 281-283): “Previously, *Vegfa* expression was documented in ECs, AT1 and AT2 cells by in situ hybridization^{16,17,24}. More recently, genetic LacZ reporters and scRNAseq analyses have reported the expression of *Vegfa* only in AT1 and ECs^{8,14,38,39}.”

New data are primarily related to a more detailed analysis of *Vegfa* isoforms which are well-validated, however, the work lacks the mechanistic studies needed to implicate these *Vegfa* isoforms in unique cell signaling, cell functions, and the specific roles of the proposed splice variants as well as their mediators. For example, regarding *hnRNPA1*, *Cpeb4*, *Elavl2*, etc. do these mediate the splicing changes? What are the functional implications of the alternatively spliced RNAs? Thus, the paper would be strengthened by data implicating potential functional roles for the *Vegfa* isoforms. For example, studies in alveolar organoids or in

other in vitro systems might be used to provide a functional readout for the Vegfa isoforms or for the proposed splicing mediators producing them.

Point 3.2 - We agree with the reviewer's suggestions. Functional experiments would greatly improve the impact of the current manuscript.

To address the concerns raised by the reviewer and test the functional specificity of VEGFA 188, as compared to VEGFA 164, we have now tested a novel hypothesis on the role of Vegfa isoform switching at perinatal stages in the lung. Since Vegfa produced by AT1 cells has been shown to induce the specification of an endothelial cell subtype, called Car4+ ECs or aerocytes (Vila Ellis et al., 2020), around the same stage as we observe the switching of Vegfa from 164 to 188, we hypothesise that VEGFA 188 may stimulate the differentiation of Car4+ ECs.

To test this, we have established E17.5 mouse lung explant cultures and stimulated them with control, mouse recombinant VEGFA164 or VEGFA188 proteins for 24h. Immunofluorescence analysis revealed that Immunofluorescence analysis revealed an increased fraction of Car4-positive ECs on explants cultured on media supplemented with VEGFA 188, as compared to those cultured on that containing VEGFA 164 or on control conditions. These results revealed a specific role of the AS isoform VEGFA 188 during lung development. These new results are presented on a **new results subsection** and on the **new Figure 7**. We have changed the **Title of the manuscript** to reflect also this new finding.

In general, the authors' conclusion is convincingly shown, and they have not overly interpreted the findings on the basis of their data. Methods and data are carefully documented, repeatable, and logically presented with attention to statistical variations.

Minor comments:

Abstract Line 22 - the alveoli contain multiple cell types, including diverse fibroblasts, pericytes, myofibroblasts as well as epithelial and endothelial cells all of which are critical for alveolar function; this might be rephrased.

Point 3.3 – We have now rephrased this sentence to clarify that the alveoli are also composed by other cell types.

Since these data add to a number of similar developmental profiles that are widely available for mouse lung, these should be referenced and discussed in light of the authors' present data, noting the similarity of their data to the published JAX laboratory bulk RNA studies and those from other consortia that are available online.

Point 3.4 - We understand reviewer's suggestion. The previously published RNAseq datasets have solely analysed gene expression changes. Thus, we agree it would be a good idea to refer these previous

datasets and compare them to ours in terms of the gene expression changes identified. We have now included this point in the current version of this manuscript.

The figures and representation were adequate.

Reviewer #3 (Significance (Required)):

Significance:

The work adds to already robust and available RNA profiles from single cell, isolated cell fractions, and whole lungs that are readily available in open access portals limiting novelty. Nevertheless, the study is well done; the novel data are modest and related to Vegfa isoforms, observations that lack functional data regarding how isoform changes during development are mediated and importantly are their unique functions of the Vegfa isoforms that might influence alveolar formation in the perinatal period? Such data would strengthen the present work and provide insights into the role of AT1-endothelial interactions around birth.

****Referee Cross-commenting****

Both reviewers have similar comments, RNA splicing is understudied and likely to add knowledge regarding lung developmental mechanism. Both reviewers suggest the need for functional data that would provide insight into the role of the proposed splicing regulators or functions of the dynamically regulated Vegf isoforms

To answer the reviewers' requests we have performed the functional experiments to address the reviewers' requests as described on **Point 3.2** regarding the function of the different VEGFA isoforms.

Due to the lack of time, staff and funds, we have not performed the experiments testing the functional impact of the splicing regulators.

References

- Bates, D.O., Cui, T.G., Doughty, J.M., Winkler, M., Sugiono, M., Shields, J.D., Peat, D., Gillatt, D., and Harper, S.J. (2002). VEGF165b, an inhibitory splice variant of vascular endothelial growth factor, is down-regulated in renal cell carcinoma. *Cancer Res.* 62, 4123–4131.
- Benjamin, J.T., Carver, B.J., Plosa, E.J., Yamamoto, Y., Miller, J.D., Liu, J.-H., van der Meer, R., Blackwell, T.S., and Prince, L.S. (2010). NF-κB Activation Limits Airway Branching through Inhibition of Sp1-Mediated Fibroblast Growth Factor-10 Expression. *J. Immunol.* 185, 4896–4903.
- Brash, J.T., Denti, L., Ruhrberg, C., and Bucher, F. (2019). VEGF188 promotes corneal reinnervation after injury. *JCI Insight* 4.
- Bridgett, S., Dellett, M., and Simpson, D.A. (2017). RNA-Sequencing data supports the existence of novel VEGFA splicing events but not of VEGFA xxx b isoforms. *Sci. Rep.* 1–11.
- Dardente, H., English, W.R., Valluru, M.K., Kanthou, C., and Simpson, D. (2020). Debunking the Myth of the

Endogenous Antiangiogenic Vegfxxx transcripts. *Trends Endocrinol. Metab.* 37, 398–409.

Harris, S., Craze, M., Newton, J., Fisher, M., Shima, D.T., and Tozer, G.M. (2012). Do Anti-Angiogenic VEGF (VEGFxxx) Isoforms Exist? A Cautionary Tale. 7, 1–14.

Jacob, A.G., and Smith, C.W.J. (2017). Intron retention as a component of regulated gene expression programs. *Hum. Genet.* 136, 1043–1057.

Mauger, O., Lemoine, F., and Scheiffele, P. (2016). Targeted Intron Retention and Excision for Rapid Gene Regulation in Response to Neuronal Activity. *Neuron* 92, 1266–1278.

Peach, C.J., Mignone, V.W., Arruda, M.A., Alcobia, D.C., Hill, S.J., Kilpatrick, L.E., and Woolard, J. (2018). Molecular pharmacology of VEGF-A isoforms: Binding and signalling at VEGFR2. *Int. J. Mol. Sci.* 19.

Prince, L.S., Okoh, V.O., Moninger, T.O., and Matalon, S. (2004). Lipopolysaccharide increases alveolar type II cell number in fetal mouse lungs through Toll-like receptor 4 and NF- κ B. *Am. J. Physiol. - Lung Cell. Mol. Physiol.* 287, 999–1006.

Ramirez, M.I., Pollack, L., Millien, G., Yu, X.C., Hinds, A., and Williams, M.C. (2002). The α -isoform of caveolin-1 is a marker of vasculogenesis in early lung development. *J. Histochem. Cytochem.* 50, 33–42.

Serra, H., Chivite, I., Angulo-Urarte, A., Soler, A., Sutherland, J.D., Arruabarrena-Aristorena, A., Ragab, A., Lim, R., Malumbres, M., Fruttiger, M., et al. (2015). PTEN mediates Notch-dependent stalk cell arrest in angiogenesis. *Nat. Commun.* 6.

Sobczak, M., Dargatz, J., and Chrzanowska-Wodnicka, M. (2010). Isolation and culture of pulmonary endothelial cells from neonatal mice. *J. Vis. Exp.* 1–4.

Vila Ellis, L., Cain, M.P., Hutchison, V., Flodby, P., Crandall, E.D., Borok, Z., Zhou, B., Ostrin, E.J., Wythe, J.D., and Chen, J. (2020). Epithelial Vegfa Specifies a Distinct Endothelial Population in the Mouse Lung. *Dev. Cell* 52, 617–630.e6.

Wang, J., Niu, N., Xu, S., and Jin, Z.G. (2019). A simple protocol for isolating mouse lung endothelial cells. *Sci. Rep.* 9, 1–10.

Yamamoto, H., Rundqvist, H., Branco, C., and Johnson, R.S. (2016). Autocrine VEGF isoforms differentially regulate endothelial cell behavior. *Front. Cell Dev. Biol.* 4, 1–12.

Yeganeh, B., Bilodeau, C., and Post, M. (2018). Explant culture for studying lung development. *Methods Mol. Biol.* 1752, 81–90.

June 27, 2022

RE: Life Science Alliance Manuscript #LSA-2022-01554R

Dr. Francisca F Vasconcelos
Instituto de Medicina Molecular João Lobo Antunes
Vascular Morphogenesis Lab
Avenida Professor Egas Moniz
Lisboa 1649-028
Portugal

Dear Dr. Vasconcelos,

Thank you for submitting your revised manuscript entitled "Aerocyte specification and lung adaptation to breathing is dependent on alternative splicing changes". We would be happy to publish your paper in Life Science Alliance pending final revisions necessary to meet our formatting guidelines.

- please address Reviewer 2's remaining comment
- please consult our manuscript preparation guidelines <https://www.life-science-alliance.org/manuscript-prep> and make sure your manuscript sections are in the correct order
- please use the [10 author names, et al.] format in your references (i.e. limit the author names to the first 10)
- please add a callout for Figure 7D and Figure S3B and Table S4 to your main manuscript text
- please add a Data Availability Statement mentioning again the accession information for the RNA-seq data

A. FINAL FILES:

B. MANUSCRIPT ORGANIZATION AND FORMATTING:

Sincerely,

Reviewer #1 (Comments to the Authors (Required)):

The authors have addressed in a satisfactory manner all my previously raised points; I believe the manuscript is stronger now and I recommend it for publication

Reviewer #2 (Comments to the Authors (Required)):

This is a revised manuscript providing mRNA profiling data of mouse lung tissue. The authors demonstrate experimental and post-natal changes in RNA splicing, a relatively understudied area of lung biology. The authors have carefully addressed all of the major concerns raised in the review by both referees and have added new data demonstrating enrichment of the Vegfa188 isoform in AT1 cells occurring concomitantly with AT1 cell differentiation and expression of Car4, a marker of aerocytes, a specialized alveolar endothelial cell. Evidence for a novel splice isoform VEGFa-i5 is provided and a subset of RNA splicing enzymes is identified that may be involved in differential splicing during alveolarization. Both reviewers requested functional data regarding splicing changes which are now provided in Figure 7. This is an important finding, thus the title change. The authors have carefully addressed and responded to major questions raised, including new data, and appropriate references. The induction of Car4 cells by VEGF188 shown in Figure 7 is an important finding but only supported by n=3 and by IF imaging of Car4. Since there is variability in explant survival in differentiation, orthogonal data e.g. RNA or other aerocyte markers supporting the role of Vegf188 in the growth or differentiation of an increased number of experimental (n's) are requested for Figure 7.

September 22, 2022

RE: Life Science Alliance Manuscript #LSA-2022-01554RR

Dr. Francisca F Vasconcelos
Instituto de Medicina Molecular João Lobo Antunes
Vascular Morphogenesis Lab
Avenida Professor Egas Moniz
Lisboa 1649-028

Dear Dr. Vasconcelos,

Thank you for submitting your Research Article entitled "Aerocyte specification and lung adaptation to breathing is dependent on alternative splicing changes". It is a pleasure to let you know that your manuscript is now accepted for publication in Life Science Alliance. Congratulations on this interesting work.

DISTRIBUTION OF MATERIALS:

Again, congratulations on a very nice paper. I hope you found the review process to be constructive and are pleased with how the manuscript was handled editorially. We look forward to future exciting submissions from your lab.

Sincerely,
